# *Caenorhabditis elegans* Dicer acts with the RIG-I-like helicase DRH-1 and RDE-4 to cleave dsRNA

**Claudia D Consalvo, Adedeji M Aderounmu†, Helen M Donelick†, P Joseph Aruscavage, Debra M Eckert, Peter S Shen\*, Brenda L Bass\***

Department of Biochemistry, University of Utah, Salt Lake City, United States

**Abstract** Invertebrates use the endoribonuclease Dicer to cleave viral dsRNA during antiviral defense, while vertebrates use RIG-I-like Receptors (RLRs), which bind viral dsRNA to trigger an interferon response. While some invertebrate Dicers act alone during antiviral defense, *Caenorhabditis elegans* Dicer acts in a complex with a dsRNA binding protein called RDE-4, and an RLR ortholog called DRH-1. We used biochemical and structural techniques to provide mechanistic insight into how these proteins function together. We found RDE-4 is important for ATP-independent and ATP-dependent cleavage reactions, while helicase domains of both DCR-1 and DRH-1 contribute to ATP-dependent cleavage. DRH-1 plays the dominant role in ATP hydrolysis, and like mammalian RLRs, has an N-terminal domain that functions in autoinhibition. A cryo-EM structure indicates DRH-1 interacts with DCR-1's helicase domain, suggesting this interaction relieves autoinhibition. Our study unravels the mechanistic basis of the collaboration between two helicases from typically distinct innate immune defense pathways.

**\*For correspondence:**
peter.shen@biochem.utah.edu (PSS);
bbass@biochem.utah.edu (BLB)

†These authors contributed equally to this work

**Competing interest:** The authors declare that no competing interests exist.

## eLife assessment

To investigate the evolutionary relationship between the RNAi pathway and innate immunity, this **valuable** study uses biochemistry and structural biology to investigate the trimeric complex of Dicer-1, DRH-1 (a RIG-I homologue), and RDE-4, which exists in *C. elegans*. The results described include rigorous kinetic analysis of the enzymatic activity of the complex and a moderate resolution cryo-EM structure. The results are **convincing** and add to the broader understanding of the evolution of antiviral defense.

## Introduction

RNA interference (RNAi) is a conserved pathway that likely had an ancestral role in defending genomes against selfish elements and viruses (*Dumesic and Madhani, 2014*; *Gutbrod and Martienssen, 2020*; *Koonin, 2017*). Modern day invertebrates still rely on RNAi for viral defense, while in most vertebrates, the interferon pathway is the primary means of fighting a viral infection. However, a role for RNAi in vertebrate innate immunity has not been ruled out (*Cullen et al., 2013*; *Poirier et al., 2021*), and as in all animals, vertebrates encode the machinery for RNAi, where it functions in regulation of endogenous gene expression via small RNAs (*Wilson and Doudna, 2013*).

A key player in RNAi is Dicer, a multidomain, ribonuclease (RNase) III family member (*Ciechanowska et al., 2021*; *Figure 1A*) that cleaves viral or endogenous dsRNA into microRNAs (miRNAs) or small interfering RNAs (siRNAs) (*Wilson and Doudna, 2013*). The tandem RNase III domains of Dicer (RNaseIIIa and RNaseIIIb) each cleave one strand to produce a staggered cleavage product, yielding a small dsRNA with 3' overhangs (3'ovrs), and strands of ~21–27 nucleotides (nts), each

with a 5′ phosphate and 3′ hydroxyl. Animal Dicers have an N-terminal helicase domain (*Figure 1A*), and whether the helicase domain functions to hydrolyze ATP coincides with antiviral functions. For example, in *Drosophila melanogaster* there are two Dicer enzymes: Dicer-1 (dmDcr1) and Dicer-2 (dmDcr2). dmDcr1 has a degenerate helicase domain, and in an ATP-independent manner, cleaves miRNA precursors to generate mature miRNA (*Jiang et al., 2005*; *Tsutsumi et al., 2011*). By contrast, ATP hydrolysis by the helicase domain of dmDcr2 fuels translocation and processive cleavage of viral or endogenous dsRNA into siRNA (*Sinha et al., 2018*; *Welker et al., 2011*), and mutations in this domain increase susceptibility to viral load (*Donelick et al., 2020*; *Marques et al., 2013*). dmDcr2 cleaves dsRNA with blunt (BLT) termini more efficiently than those with 3′ovrs, and it is speculated that this allows discrimination between self and nonself dsRNA (*Singh et al., 2021*; *Sinha et al., 2018*; *Sinha et al., 2015*; *Welker et al., 2011*). Despite having a conserved helicase domain, a requirement for ATP has not been reported for vertebrate Dicers, including human Dicer (hsDcr-1); in fact, the helicase domain of hsDcr-1 inhibits cleavage activity (*Ma et al., 2008*).

While Dicer binds and cleaves viral dsRNA to trigger antiviral RNAi in invertebrates, the recognition of viral dsRNA during a vertebrate interferon response is performed by RIG-I-like Receptors (RLRs) (*Ahmad and Hur, 2015*). Despite differences in their domain organization and functions, Dicer and RLRs share a conserved DExD/H-box helicase domain (*Figure 1A*, *Luo et al., 2013*). The best characterized RLRs, RIG-I and MDA5, use their ATP-dependent helicase domains to bind viral dsRNA and stimulate a cascade of events that induces the Type I Interferon response (*Kowalinski et al., 2011*; *Luo et al., 2011*; *Rehwinkel and Gack, 2020*). Like dmDcr2, RIG-I distinguishes between self from non-self using dsRNA termini by recognizing BLT dsRNA; in contrast, MDA5 prefers long dsRNA (*Peisley et al., 2011*; *Schlee et al., 2009*). Interestingly, in some cases, Dicer and the interferon response have antagonistic interactions, for example, the RLR LGP2 inhibits human Dicer's ability to cleave dsRNA (*van der Veen et al., 2018*).

The conservation of Dicer's helicase domain with that of RLRs suggests this domain functioned in antiviral defense in a common ancestor, and it is fascinating to imagine the host-virus arms race that led to the distinctly different innate immune pathways of extant vertebrates and invertebrates. *C. elegans* is of keen interest in this regard because its single encoded Dicer functions in complex with an RLR homolog to mediate an antiviral RNAi response (*Ashe et al., 2013*; *Guo et al., 2013*; *Thivierge et al., 2012*). The antiviral complex in *C. elegans* includes Dicer (DCR-1), the RIG-I ortholog called Dicer-Related Helicase-1 (DRH-1), and a dsRNA binding protein called RNAi deficient-4 (RDE-4; *Figure 1A*, *Tabara et al., 2002*; *Thivierge et al., 2012*). Like RIG-I and MDA5 (*Ahmad and Hur, 2015*; *Guo et al., 2013*), DRH-1 has an N-terminal domain (NTD), a central helicase domain, and a C-terminal regulatory domain (CTD; *Figure 1A*). RDE-4 is a dsRNA binding protein (dsRBP) that contains three dsRNA binding motifs and has a high affinity for binding long dsRNA (*Parker et al., 2008*; *Parker et al., 2006*).

RNAi was first discovered in *C. elegans* (*Fire et al., 1998*), but efforts to characterize DCR-1 have been hampered by challenges to purify it in a recombinant form. Here, we overcome this bottleneck by co-expressing and purifying the antiviral complex comprising DCR-1, DRH-1, and RDE-4. The complex exhibits some features previously observed in vitro for purified dmDcr2, such as cleavage and ATP hydrolysis activity that depends on dsRNA termini (*Cenik et al., 2011*; *Singh et al., 2021*; *Sinha et al., 2018*). However, comparison of the activity of the wildtype antiviral complex with complexes containing point mutations in the helicase domain of DCR-1 or DRH-1 demonstrate that ATP hydrolysis activity comes primarily from DRH-1. While Dicer enzymes are commonly found to have dsRBPs that bind their helicase domain to facilitate substrate specificity and regulate cleavage (*Hansen et al., 2019*), RDE-4 plays a critical role in all activities of the *C. elegans* antiviral complex, and dsRNA binding, cleavage and ATP hydrolysis are all greatly compromised in its absence. Finally, we supplement our biochemical studies with cryo-EM structure determination, providing mechanistic insight into how these three proteins cooperate in antiviral defense.

## Results
### Purification of the *C. elegans* antiviral complex, DCR-1•DRH-1•RDE-4

We co-expressed DCR-1, DRH-1, and RDE-4 in Sf9 cells from a single baculovirus expression plasmid (*Weissmann et al., 2016*). The construct included a One-STrEP-FLAG (OSF) tag on the N-terminus of

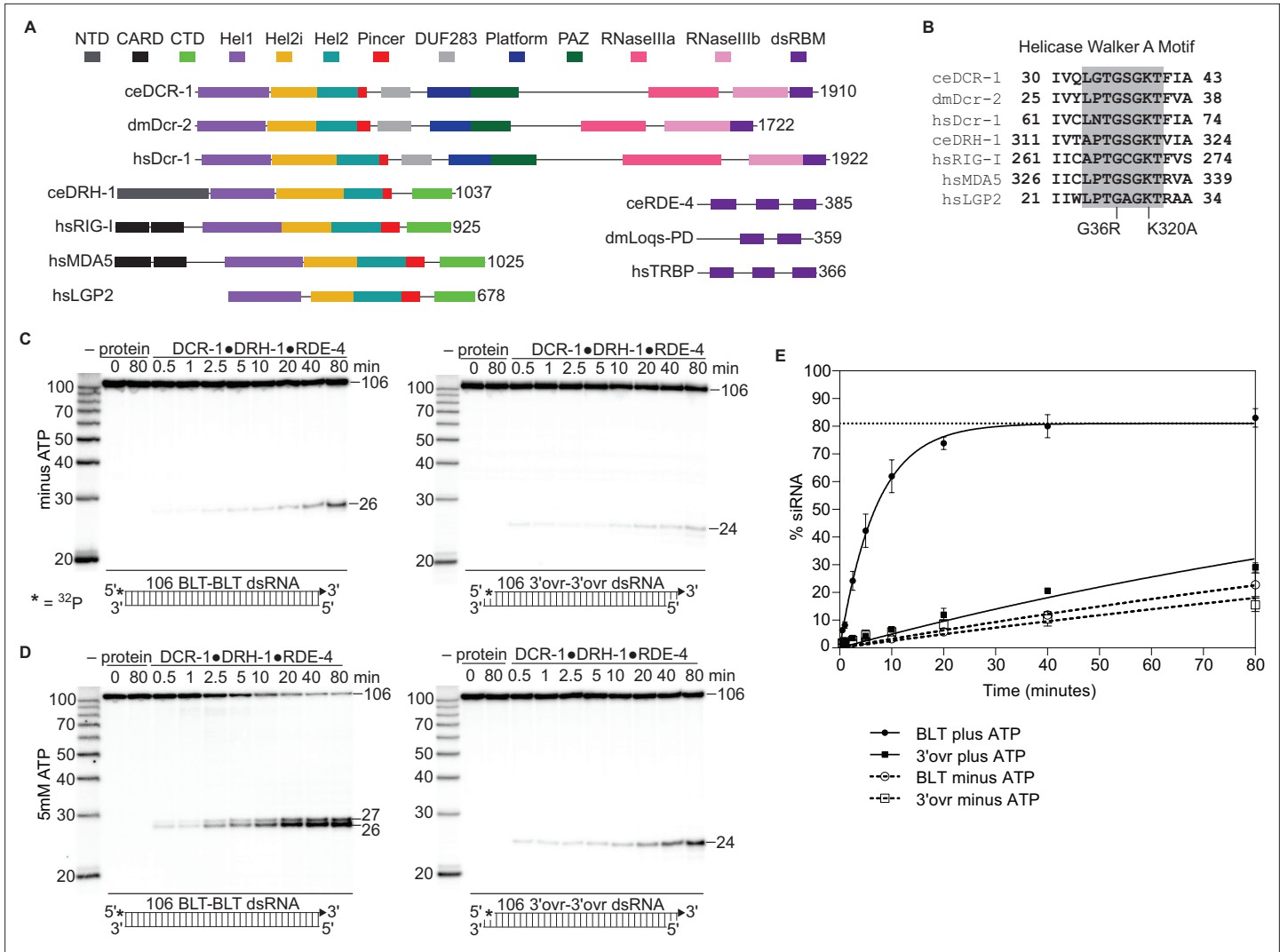

**Figure 1.** The *C. elegans* antiviral complex DCR-1•DRH-1•RDE-4 preferentially cleaves blunt dsRNA in an ATP-dependent manner. (**A**) Colored rectangles depict conserved domains. NCBI conserved domains (*Sayers et al., 2011*), Clustal Omega Multiple Sequence Alignments (*Goujon et al., 2010*; *McWilliam et al., 2013*; *Sievers et al., 2011*), and previous reports were used in defining domain boundaries (*Guo et al., 2013*; *Hansen et al., 2019*; *Sinha et al., 2018*). Numbers to right of open-reading frame indicate total amino acids in protein. Organism is indicated: ce, *C. elegans*; dm, *Drosophila melanogaster*, hs, *Homo sapiens*. (**B**) Amino acids within Walker A motif (shaded) of the Hel1 subdomain with mutations indicated for DCR-1 (G36R) and DRH-1 (K320A). Number indicates the residues at the beginning and end. (**C**) Single-turnover cleavage assays of DCR-1•DRH-1•RDE-4 (25 nM) with 106 BLT or 3'ovr dsRNA (1 nM) at 20 °C in the absence of ATP. Sense (top) strand was 5' $^{32}$P-end labeled (*) and contained a 2',3' cyclic phosphate (filled triangle in cartoon under gel) to minimize cleavage from that end (see *Figure 1—figure supplement 2*). Products were separated by 17% denaturing PAGE, and a representative PhosphorImage is shown (n≥3). Left, marker nucleotide lengths. See also *Figure 1—figure supplements 1–3*. (**D**) Same as C except with 5 mM ATP added to the reaction. (**E**) Quantification of single-turnover assays as in C and D with symbol key below. Data points are mean ± SD (n=3). Dotted line indicates amplitude constraint (see Materials and methods).

The online version of this article includes the following source data and figure supplement(s) for figure 1:

**Source data 1.** Raw digital image of cleavage phosphorimager plate used in *Figure 1C*, left panel.

**Source data 2.** Raw digital image of cleavage phosphorimager plate used in *Figure 1C*, right panel.

**Source data 3.** Raw digital image of cleavage phosphorimager plate used in *Figure 1D*, left panel.

**Source data 4.** Raw digital image of cleavage phosphorimager plate used in *Figure 1D*, right panel.

**Figure supplement 1.** Analysis and validation of *C. elegans* antiviral complex after gel filtration.

**Figure supplement 1—source data 1.** Raw digital image of SDS-PAGE gel used in *Figure 1—figure supplement 1A*.

**Figure supplement 1—source data 2.** Raw digital image of western blot used in *Figure 1—figure supplement 1B*, left panel.

**Figure supplement 1—source data 3.** Raw digital image of western blot used in *Figure 1—figure supplement 1B*, middle panel.

*Figure 1 continued on next page*

*Figure 1 continued*

**Figure supplement 1—source data 4.** Raw digital image of western blot used in *Figure 1—figure supplement 1B*, right panel.

**Figure supplement 1—source data 5.** Raw digital image of SDS-Page gel used in *Figure 1—figure supplement 1D*.

**Figure supplement 2.** 2′,3′ Cyclic phosphates block ATP-independent cleavage and reveal an ATP-dependent side reaction characterized by heterogeneous products.

**Figure supplement 2—source data 1.** Raw digital image of cleavage phosphorimager plate used in *Figure 1—figure supplement 2A*.

**Figure supplement 2—source data 2.** Raw digital image of cleavage phosphorimager plate used in *Figure 1—figure supplement 2B*.

**Figure supplement 3.** DCR-1•DRH-1•RDE-4 cleaves blunt dsRNA in an ATP-independent manner.

**Figure supplement 3—source data 1.** Raw digital image of cleavage phosphorimager plate used in *Figure 1—figure supplement 3A*.

**Figure supplement 3—source data 2.** Raw digital image of cleavage phosphorimager plate used in *Figure 1—figure supplement 3B*.

**Figure supplement 3—source data 3.** Raw digital image of cleavage phosphorimager plate used in *Figure 1—figure supplement 3D*.

DRH-1, allowing affinity purification using Strep-Tactin, and subsequently the complex was purified to ≥90% homogeneity by size-exclusion chromatography (SEC). Analysis of the complex by SDS-PAGE showed three major bands (*Figure 1—figure supplement 1A*), which were validated as DCR-1, DRH-1 and RDE-4, using mass spectrometry and western blot analyses (*Figure 1—figure supplement 1B and C*). Using this approach, we also purified three variants of the complex containing point mutations in the Walker A motif of DCR-1 (G36R), DRH-1 (K320A), or lacking RDE-4 (*Figure 1B*, *Figure 1—figure supplement 1D*). Sedimentation velocity analytical ultracentrifugation (SV-AUC) analyses indicated DCR-1•DRH-1•RDE-4 and DCR-1•DRH-1 complexes have a 1:1:1 and 1:1 stoichiometry, respectively (*Figure 1—figure supplement 1E–G*).

## The catalytically active DCR-1•DRH-1•RDE-4 complex cleaves dsRNA in a terminus and ATP-dependent manner

Like *C. elegans*, *D. melanogaster* uses RNAi for antiviral defense (*Donelick et al., 2020*; *Marques et al., 2013*), and the key enzyme in this pathway, dmDcr2, is biochemically well characterized (*Cenik et al., 2011*; *Singh et al., 2021*; *Sinha et al., 2018*). Although the antiviral complex contains two proteins in addition to DCR-1, as a first step in characterizing its biochemical activity, we tested whether the *C. elegans* antiviral complex had activities exhibited by dmDcr2 alone, such as dependencies on ATP and dsRNA termini. Using the *C. elegans* antiviral complex, we conducted single-turnover cleavage assays using a 106 base-pair (bp) dsRNA (106-dsRNA) with BLT or 3′ovr termini either without or with ATP (*Figure 1C and D*). The sense strand was 5′ $^{32}$P-end-labeled and contained a 2′,3′ cyclic phosphate at the opposite end to minimize cleavage from that end (*Figure 1—figure supplement 2*), allowing us to focus on the first cleavage event from the radiolabeled terminus (*Figure 1C and D*).

In the absence of ATP, the complex inefficiently cleaved BLT and 3′ovr dsRNA at similar rates (*Figure 1C* and dashed lines in *Figure 1E*), but the radiolabeled siRNA product was 26nt using a BLT dsRNA, and 24nt using 3′ovr dsRNA. With ATP, the efficiency of BLT dsRNA cleavage was dramatically enhanced, and gave rise to an additional ATP-dependent siRNA product of 27nt, while the time course of 3′ovr dsRNA cleavage was relatively unchanged (*Figure 1D and E*). The 26nt and 27nt siRNAs produced from a BLT dsRNA terminus were consistent with previous experiments using *C. elegans* embryo extracts (*Ruby et al., 2006*; *Welker et al., 2011*). However, the 24nt siRNA band with 3′ovr dsRNA was one nucleotide longer than previously reported (*Welker et al., 2011*), possibly due to a protein cofactor that is not present in our in vitro system.

Differential processing of dsRNA based on termini is reminiscent of dmDcr2, which preferentially cleaves BLT dsRNA in the presence of ATP, even without other factors (*Sinha et al., 2018*). However, there are notable differences between the DCR-1•DRH-1•RDE-4 complex and dmDcr2. dmDcr2 cannot cleave BLT dsRNA without ATP (*Sinha et al., 2015*), but the DCR-1•DRH-1•RDE-4 complex readily cleaved dsRNA in the absence of ATP (*Figure 1C*, *Figure 1—figure supplement 3A*). In addition, with ATP, at least in vitro, dmDcr2 threads dsRNA through the helicase domain, and as dsRNA passes the RNase III active sites, it is cleaved to yield heterogeneous-sized siRNA products (*Sinha et al., 2018*). In contrast, we observed that the antiviral complex produced siRNAs of discrete lengths. Possibly, the presence of DRH-1 and RDE-4 allow a more precise measuring of siRNA products.

DRH-1 is an RLR homolog, and RIG-I primarily recognizes BLT dsRNA with a 5′-triphosphate (5′ppp). By contrast, with the antiviral complex we saw little difference in cleavage (*Figure 1—figure supplement 3B and C*) or ATP hydrolysis (*Figure 1—figure supplement 3D and E*) when comparing BLT 106-dsRNA with a 5′p or 5′ppp. Thus, for the antiviral complex, a BLT terminus is more important for recognition than the phosphorylation state, at least under these conditions. This is also the case for dmDcr2, where 5′ phosphorylation state does not impact its ability to cleave dsRNA (*Sinha et al., 2018*).

## ATP-dependent cleavage is mediated by DCR-1 and DRH-1 helicase domains, whereas RDE-4 is needed for both ATP-independent and ATP-dependent cleavage

To understand contributions of each protein to terminus-dependent processing of dsRNA, we compared cleavage reactions of wildtype antiviral complex to the three variants, in the presence or absence of ATP, or with the nonhydrolyzable analog ATPγS (*Figure 2A and B*). Cleavage efficiency using the wildtype complex (*Figure 2A and B*, lanes 1–3) increased with ATP, but not with ATPγS, indicating that optimal cleavage requires ATP hydrolysis. The 27nt siRNA produced from BLT dsRNA was observed with ATP, but not with ATPγS (*Figure 2A*, lanes 2 and 3), emphasizing that this product is completely dependent on ATP hydrolysis. Again, in the presence of ATP, the reaction with BLT dsRNA ($k_{obs}$, 0.14 min$^{-1}$) was more efficient compared to that with 3′ovr dsRNA ($k_{obs}$, 0.006 min$^{-1}$; *Figure 2C* and *Table 1*).

In the presence of ATP, antiviral complexes with a point mutation in the Walker A motif (*Figure 1B*) of the helicase domain of DCR-1 (G36R) or DRH-1 (K320A), reduced cleavage efficiency of BLT dsRNA, demonstrating that ATP hydrolysis by both helicase domains is important for optimal cleavage from this terminus (*Figure 2A*, compare lanes 3, 6 and 9, *Figure 2C*, compare solid black, green and blue curves, *Table 1*, *Figure 2—figure supplement 1A and B*). The complex containing the DCR-1 G36R mutation showed a loss of the 27nt siRNA band (*Figure 2A*, compare lanes 3 and 6, *Figure 2—figure supplement 1A*). This is consistent with work done using extracts from *C. elegans* containing a mutation in DCR-1′s helicase domain (*Welker et al., 2011*), thereby confirming a role for DCR-1′s Walker A motif in production of this slightly longer siRNA. By contrast, the DRH-1 K320A mutation reduced overall cleavage (*Figure 2C*, *Table 1*), but the siRNA doublet was apparent (*Figure 2A*, compare lane 3 with 9; *Figure 2—figure supplement 1B*). For the Walker A mutations in either DCR-1 or DRH-1, we observed no effect on cleavage of BLT or 3′ovr dsRNA in the absence of ATP or with ATPγS (*Figure 2A and B*, compare lanes 1–2 with 4–5 and 7–8). With 3′ovr dsRNA, we observed little effect of the Walker A mutations on cleavage, regardless of the presence or absence of nucleotide (*Figure 2B*, compare lanes 1–9, *Figure 2C*, dashed curves, *Table 1*, *Figure 2—figure supplement 1A and B*).

The most dramatic effects among the three variants were observed with the DCR-1•DRH-1 complex that lacked the RDE-4 protein. Cleavage of BLT dsRNA was only observed with ATP, and this occurred with low efficiency (*Figure 2A*, compare lanes 1–3 with lanes 10–12, *Figure 2C*, compare solid black and purple curves, *Figure 2—figure supplement 1C*). Cleavage with this complex was undetectable with 3′ovr dsRNA regardless of whether nucleotide was present (*Figure 2B*, compare lanes 1–3 with 10–12, *Figure 2—figure supplement 1C*). While RDE-4 was clearly necessary for 3′ovr cleavage, the small amount of siRNA of the correct size observed with BLT dsRNA and ATP indicated measuring and cleavage by the RNase III domains were functional in the DCR-1•DRH-1 complex.

Taken together, these studies indicate RDE-4 plays a significant role in promoting all cleavage reactions by the DCR-1•DRH-1•RDE-4 complex, whereas the helicase domains of DCR-1 and DRH-1 only impact ATP hydrolysis-dependent cleavage, and this effect was most observable with BLT dsRNA (*Figure 2A–C*). The loss in cleavage efficiency in the presence of ATP was greater with the DCR-1•DRH-1$^{K320A}$•RDE-4 complex (BLT, $k_{obs}$, 0.013 min$^{-1}$) than the DCR-1$^{G36R}$•DRH-1•RDE-4 complex (BLT, $k_{obs}$, 0.05 min$^{-1}$; *Table 1*), albeit the loss of the 27nt ATP dependent band was only observed with the latter, suggesting that DCR-1 is involved in an ATP hydrolysis-dependent measuring step.

## DRH-1 is essential for in vitro ATP hydrolysis in the *C. elegans* antiviral complex

Dicer and RLRs belong to the Super Family 2 nucleic acid-dependent ATPases (SF2 helicases), which require dsRNA binding for optimal enzymatic activity (*Luo et al., 2013*). As expected,

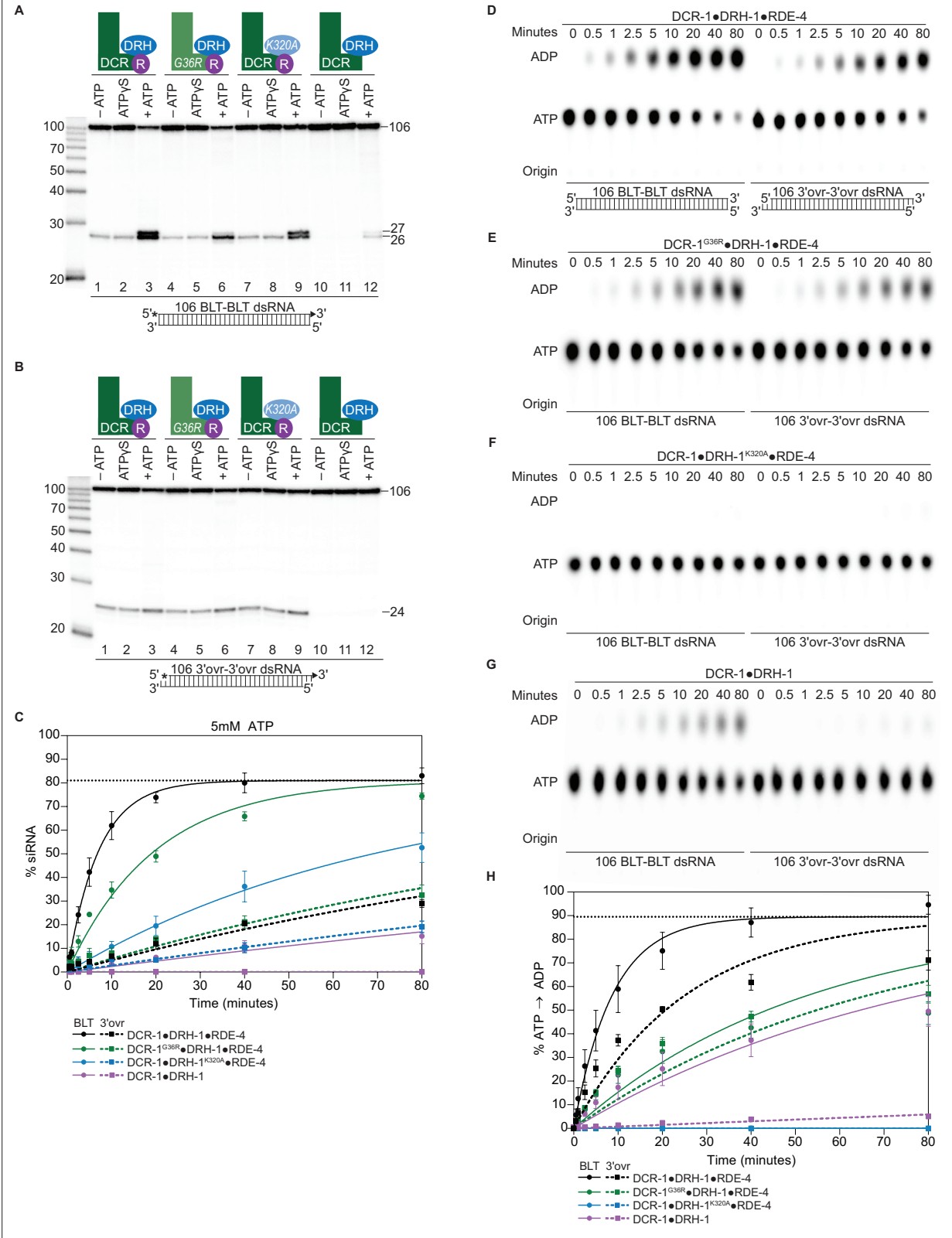

**Figure 2.** ATP-dependent cleavage reactions are mediated by DCR-1 and DRH-1 helicase domains, whereas RDE-4 is needed for ATP-independent and ATP-dependent cleavage reactions. (**A**) Cartoons indicate complexes and variants, with mutations in DCR-1 (green) and DRH-1 (blue) indicated with the amino acid change, and the presence of RDE-4 (R) represented with a purple circle. Single-turnover cleavage assays of 106 BLT dsRNA (1 nM) with indicated protein complex (25 nM) ±5 mM ATP or ATPγS, incubated at 20 °C for 60 min. Sense strand was 5′ $^{32}$P-end labeled (*) and contained

*Figure 2 continued on next page*

*Figure 2 continued*

a 2′,3′ cyclic phosphate (filled triangle in cartoon under gel) to minimize cleavage from the opposite end (see *Figure 1—figure supplement 2*). Products were separated by 17% denaturing PAGE, and a representative PhosphorImage is shown (n=3). Left, marker nucleotide lengths. (**B**) Same as A except with 106 3′ovr dsRNA (1 nM). (**C**) Quantification of single-turnover assays over time, using wildtype or variant antiviral complexes as indicated (25 nM), with 106 BLT dsRNA (1 nM) or 106 3′ovr dsRNA (1 nM) plus 5 mM ATP. Data points are mean ± SD (n=3). See *Figure 2—figure supplement 1A-C* for representative primary data. Plateau constrained to 81.02 (dotted line). See Materials and methods. (**D**) DCR-1•DRH-1•RDE-4 (25 nM) was incubated with 106 BLT or 3′ovr dsRNA (400 nM) with 100 μM α-$^{32}$P-ATP at 20 °C. ATP hydrolysis monitored by thin-layer chromatography (TLC), and a representative PhosphorImage is shown (n≥3). Positions of origin, ATP, and ADP are indicated. See also *Figure 2—figure supplement 1D and E*. (**E–G**) Same as D except with complexes indicated. (**H**) Quantification of ATP hydrolysis assays as in D – G. Data points are mean ± SD (n=3) with symbol key below graph. Plateau constrained to 89.55 (dotted line). See Materials and methods.

The online version of this article includes the following source data and figure supplement(s) for figure 2:

**Source data 1.** Raw digital image of cleavage phosphorimager plate used in *Figure 2A*.

**Source data 2.** Raw digital image of cleavage phosphorimager plate used in *Figure 2B*.

**Source data 3.** Raw digital image of thin-layer chromatography plate used in *Figure 2D*, left panel.

**Source data 4.** Raw digital image of thin-layer chromatography plate used in *Figure 2D*, right panel.

**Source data 5.** Raw digital image of thin-layer chromatography plate used in *Figure 2E*.

**Source data 6.** Raw digital image of thin-layer chromatography plate used in *Figure 2F*.

**Source data 7.** Raw digital image of thin-layer chromatography plate used in *Figure 2G*.

**Figure supplement 1.** DCR-1, DRH-1, and RDE-4 affect ATP-dependent cleavage rates.

**Figure supplement 1—source data 1.** Raw digital image of cleavage phosphorimager plate used in *Figure 2—figure supplement 1A*, left panel.

**Figure supplement 1—source data 2.** Raw digital image of cleavage phosphorimager plate used in *Figure 2—figure supplement 1A*, right panel.

**Figure supplement 1—source data 3.** Raw digital image of cleavage phosphorimager plate used in *Figure 2—figure supplement 1B*, left panel.

**Figure supplement 1—source data 4.** Raw digital image of cleavage phosphorimager plate used in *Figure 2—figure supplement 1B*, right panel.

**Figure supplement 1—source data 5.** Raw digital image of cleavage phosphorimager plate used in *Figure 2—figure supplement 1C*, left panel.

**Figure supplement 1—source data 6.** Raw digital image of cleavage phosphorimager plate used in *Figure 2—figure supplement 1C*, right panel.

**Figure supplement 1—source data 7.** Raw digital image of thin-layer chromatography plate used in *Figure 2—figure supplement 1D*.

DCR-1•DRH-1•RDE-4 showed negligible amounts of ATP hydrolysis in the absence of dsRNA or in the presence of single-stranded RNA (*Figure 2—figure supplement 1D and E*) but had robust ATP hydrolysis activity in the presence of dsRNA (*Figure 2D and H*). Our cleavage assays indicated that ATP hydrolysis by both helicases affected cleavage (*Figure 2A and C*). In order to gain more insight into the ATP-dependent roles of DCR-1 and DRH-1, we directly monitored conversion of ATP to ADP over time, in the presence of BLT or 3′ovr dsRNA, for the wildtype antiviral complex and all variants (*Figure 2D–H*; *Table 1*). Reactions were performed with excess dsRNA to focus on differences in the catalytic step, rather than differences in affinity for dsRNA, and products were separated by Thin-Layer Chromatography (TLC). Again, similarities to dmDcr2 were observed; for example, the wildtype complex was more efficient in ATP hydrolysis in the presence of BLT dsRNA (BLT, $k_{obs}$, 0.11 min$^{-1}$) compared to 3′ovr dsRNA (3′ovr, $k_{obs}$, 0.03 min$^{-1}$; *Table 1*, *Figure 2D and H*). Unexpectedly, while reaction with the complex containing DCR-1$^{G36R}$ reduced the rate of ATP hydrolysis for both dsRNA termini (*Figure 2E and H*), ATP hydrolysis was completely eliminated with complexes containing DRH-1$^{K320A}$ (*Figure 2F and H*). The DCR-1•DRH-1 variant that lacked RDE-4 was also severely compromised in the production of ADP over time (*Figure 2G and H*).

## Proteins in the antiviral complex contribute to high affinity binding to dsRNA

We used gel-shift assays to compare dsRNA binding by the wildtype antiviral complex with the three variants. The wildtype antiviral complex and all variants bound 106-dsRNA tightly, showing dissociation constants (K$_d$s) ranging from 0.3 to 29 nM (*Figure 3*, *Figure 3—figure supplement 1*, *Table 1*, and *Supplementary file 1a*), with only small differences in the presence or absence of ATP or with different termini.

Despite the large mass of the antiviral complex (386 kDa), distinct, but sometimes diffuse, gel shifts were observed. The DCR-1•DRH-1•RDE-4 wildtype complex showed a similar gel-shift pattern for BLT and 3′ovr dsRNA (*Figure 3A*, *Figure 3—figure supplement 1A*), with two prominent bands in the

**Table 1.** Summary of $k_{obs}$, $t_{1/2}$, and $K_d$ values.

| | $k_{obs}$ (min⁻¹) (Cleavage) | $t_{1/2}$ (min) (Cleavage) | $k_{obs}$ (min⁻¹) (ATP Hydrolysis) | $K_d$ (nM) | |
|---|---|---|---|---|---|
| | +ATP | +ATP | +ATP | - ATP | +ATP |
| **DCR-1•DRH-1•RDE-4** | | | | | |
| 106 BLT | 0.14±0.01 | 4.9 | 0.11±0.01 | 0.30±0.04 | 0.53±0.04 |
| 106 3'ovr | 0.006±0.0005 | 109.6 | 0.03±0.005 | 0.37±0.04 | 0.87±0.10 |
| **DCR-1^G36R•DRH-1•RDE-4** | | | | | |
| 106 BLT | 0.05±0.004 | 13.65 | 0.018±0.002 | 0.42±0.06 | 0.53±0.05 |
| 106 3'ovr | 0.007±0.0007 | 95.99 | 0.014±0.002 | 0.58±0.07 | 0.85±0.08 |
| **DCR-1•DRH-1^K320A•RDE-4** | | | | | |
| 106 BLT | 0.013±0.001 | 49.77 | n.d. | 1.12±0.12 | 1.31±0.13 |
| 106 3'ovr | 0.003±0.0004 | 199.6 | n.d. | 0.99±0.10 | 1.60±0.15 |
| **DCR-1•DRH-1** | | | | | |
| 106 BLT | 0.002±0.0003 | 235.2 | 0.012±0.001 | 2.11±0.28 | 5.18±0.33 |
| 106 3'ovr | n.d. | n.d. | n.d. | 7.50±0.80 | 28.64±2.03 |
| **ΔNTD DRH-1** | | | | | |
| 106 BLT | | | 0.08 | | |
| 106 3'ovr | | | 0.002 | | |

Values shown are mean ± SD (n=3). n.d., not detected.

absence of ATP that became diffuse in the presence of ATP. These diffuse bands were not observed when ATPγS was substituted for ATP (*Figure 3—figure supplement 2A*), suggesting ATP hydrolysis promotes the diffuse appearance. These general trends were observed for gel-shift patterns using complexes containing point mutations in the helicase domain of DCR-1 or DRH-1 (*Figure 3B and C*, *Figure 3—figure supplement 1B and C*), but consistent with its dominant role in ATP hydrolysis, the addition of ATP to DCR-1•DRH-1^K320A•RDE-4 did not lead to more diffuse bands (*Figure 3C*, *Figure 3—figure supplement 1C*). The two-band gel shifts observed with wildtype and helicase mutant complexes were not observed with the DCR-1•DRH-1 complex that lacked RDE-4 (*Figure 3D*, *Figure 3—figure supplement 1D*). This complex showed a diffuse gel-shift, suggesting the presence of RDE-4 is required for the two discrete bands observed with complexes that contained all three proteins.

Quantification of multiple gel-shift experiments allowed determination of dissociation constants (*Table 1* and *Supplementary file 1a*). Binding affinities under the different conditions were similar for DCR-1•DRH-1•RDE-4 and the DCR-1^G36R•DRH-1•RDE-4 complex, while affinities decreased two- to threeold for the DCR-1•DRH-1^K320A•RDE-4 complex. The DCR-1•DRH-1 complex that lacked RDE-4, while still binding tightly, was 7- to 30-fold weaker than the wildtype complex, depending on the condition. For all complexes, addition of ATP decreased binding affinity. For complexes that contained all three proteins, BLT dsRNA was bound slightly tighter than 3'ovr dsRNA in the presence of ATP, but terminus dependence was not observed without ATP. The complex lacking RDE-4 showed the greatest differences in ATP and terminus dependence. Adding ATPγS instead of ATP decreased binding affinity further than ATP did, suggesting that ATP binding, not hydrolysis, is the primary force driving the loss of binding affinity (*Figure 3—figure supplement 2A, B, and G*, *Supplementary file 1a*).

Studies of hsDcr-1 and dmDcr2 indicate ATP-independent cleavage is mediated by the Platform•PAZ domain, rather than the helicase domain (*Park et al., 2011*; *Singh et al., 2021*; *Sinha et al., 2018*; *Tian et al., 2014*). Our observation that ATP-independent cleavage is lost when dsRNA termini are blocked with 2′, 3′-cyclic phosphates (*Figure 1—figure supplement 2*), suggests this modification precludes interaction of dsRNA with the Platform•PAZ domain of DCR-1. To delineate contributions of

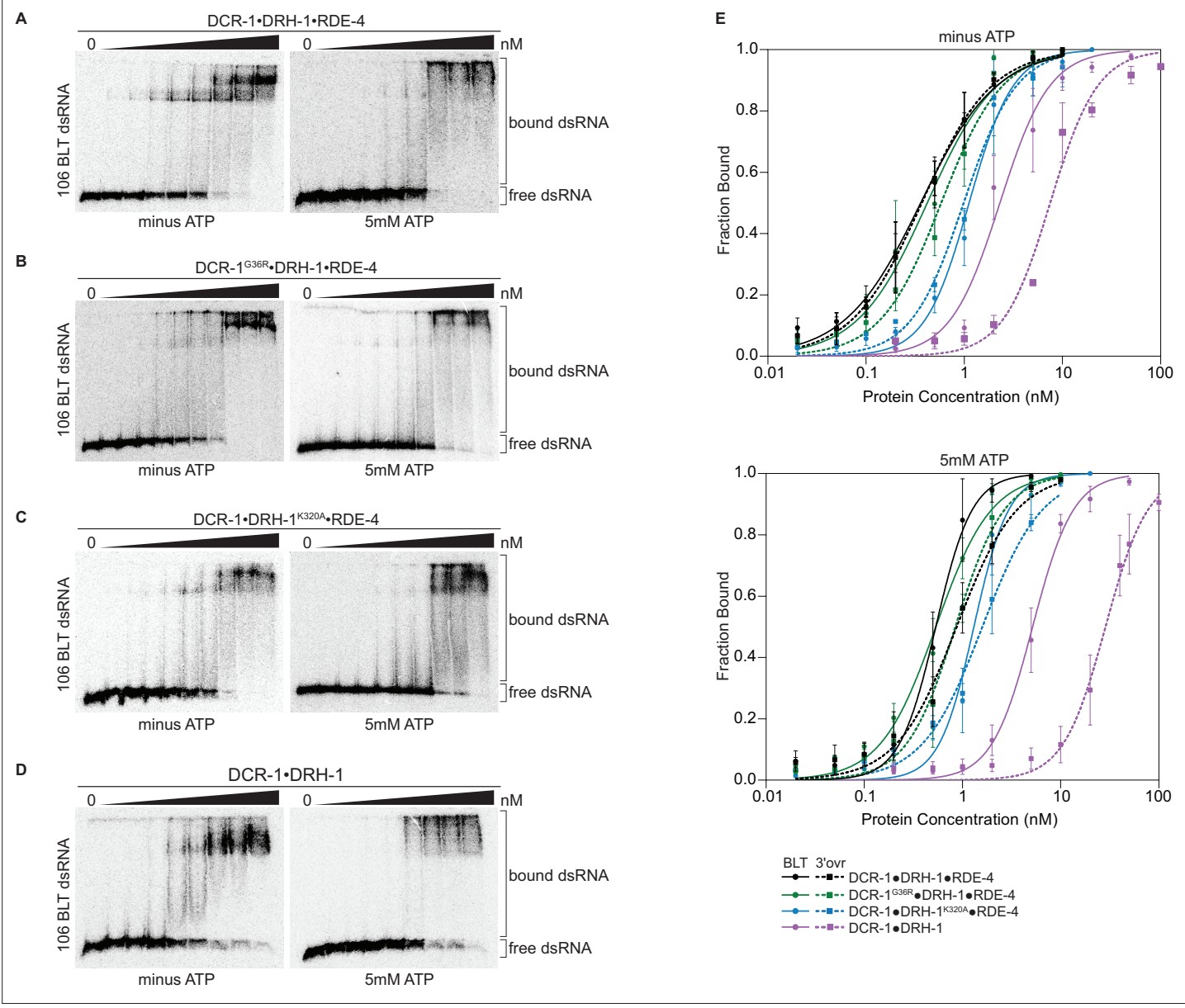

**Figure 3.** Binding affinity of DCR-1•DRH-1•RDE-4 wildtype and mutant complexes for 106 BLT dsRNA in the absence or presence of ATP.
(**A**) Representative PhosphorImages showing gel mobility shift assays with increasing concentrations ranging from 0 to 5 nM of DCR-1•DRH-1•RDE-4 with 106 BLT dsRNA ±5 mM ATP as indicated. Sense strand was 5′ $^{32}$P-end labeled (*) and contained a 2′,3′ cyclic phosphate. As labeled on right, all dsRNA that migrated through the gel more slowly than dsRNA$_{free}$ was considered bound. (**B–D**) Same as A except with indicated complexes. Protein concentrations increased from left to right as indicated and range from 0 to 10 nM (**B** and **C**), and 0–50 nM (**D**). (**E**) Radioactivity in PhosphorImages as in A – D was quantified to generate binding isotherms for wildtype and mutant complexes ±5 mM ATP (see key). Fraction bound was determined using radioactivity for dsRNA$_{free}$ and dsRNA$_{bound}$. Data was fit to calculate dissociation constant, $K_d$, using the Hill formalism, where fraction bound = $1/(1 + (K_d^n/[P]^n))$. Data points, mean ± SD (n≥3). See also ***Figure 3—figure supplement 1***.

The online version of this article includes the following source data and figure supplement(s) for figure 3:

**Source data 1.** Raw digital image of gel shift phosphorimager plate used in ***Figure 3A***, left panel.

**Source data 2.** Raw digital image of gel shift phosphorimager plate used in ***Figure 3A***, right panel.

**Source data 3.** Raw digital image of gel shift phosphorimager plate used in ***Figure 3B***, left panel.

**Source data 4.** Raw digital image of gel shift phosphorimager plate used in ***Figure 3B***, right panel.

**Source data 5.** Raw digital image of gel shift phosphorimager plate used in ***Figure 3C***, left panel.

**Source data 6.** Raw digital image of gel shift phosphorimager plate used in ***Figure 3C***, right panel.

*Figure 3 continued on next page*

*Figure 3 continued*

**Source data 7.** Raw digital image of gel shift phosphorimager plate used in *Figure 3D*, left panel.

**Source data 8.** Raw digital image of gel shift phosphorimager plate used in *Figure 3D*, right panel.

**Figure supplement 1.** Binding affinity of DCR-1•DRH-1•RDE-4 wildtype and mutant complexes for 106 3'ovr dsRNA in the absence or presence of ATP.

**Figure supplement 1—source data 1.** Raw digital image of gel shift phosphorimager plate used in *Figure 3—figure supplement 1A*, left panel.

**Figure supplement 1—source data 2.** Raw digital image of gel shift phosphorimager plate used in *Figure 3—figure supplement 1A*, right panel.

**Figure supplement 1—source data 3.** Raw digital image of gel shift phosphorimager plate used in *Figure 3—figure supplement 1B*, left panel.

**Figure supplement 1—source data 4.** Raw digital image of gel shift phosphorimager plate used in *Figure 3—figure supplement 1B*, right panel.

**Figure supplement 1—source data 5.** Raw digital image of gel shift phosphorimager plate used in *Figure 3—figure supplement 1C*, left panel.

**Figure supplement 1—source data 6.** Raw digital image of gel shift phosphorimager plate used in *Figure 3—figure supplement 1C*, right panel.

**Figure supplement 1—source data 7.** Raw digital image of gel shift phosphorimager plate used in *Figure 3—figure supplement 1D*, left panel.

**Figure supplement 1—source data 8.** Raw digital image of gel shift phosphorimager plate used in *Figure 3—figure supplement 1D*, right panel.

**Figure supplement 2.** Binding affinity of DCR-1•DRH-1•RDE-4 and DCR-1•DRH-1 for 106 BLT dsRNA in the presence of ATPγS, and for 106 BLT blocked with cyclic phosphate on both ends in the presence and absence of ATP.

**Figure supplement 2—source data 1.** Raw digital image of gel shift phosphorimager plate used in *Figure 3—figure supplement 2A*.

**Figure supplement 2—source data 2.** Raw digital image of gel shift phosphorimager plate used in *Figure 3—figure supplement 2B*.

**Figure supplement 2—source data 3.** Raw digital image of gel shift phosphorimager plate used in *Figure 3—figure supplement 2C*.

**Figure supplement 2—source data 4.** Raw digital image of gel shift phosphorimager plate used in *Figure 3—figure supplement 2D*.

**Figure supplement 2—source data 5.** Raw digital image of gel shift phosphorimager plate used in *Figure 3—figure supplement 2E*.

**Figure supplement 2—source data 6.** Raw digital image of gel shift phosphorimager plate used in *Figure 3—figure supplement 2F*.

binding to the Platform•PAZ domain of DCR-1, we performed binding assays of DCR-1•DRH-1•RDE-4 and a 106nt BLT dsRNA with 2', 3'-cyclic phosphates on both ends. With ATP, binding of DCR-1•DRH-1•RDE-4 or DCR-1•DRH-1 to cyclic phosphate-blocked BLT dsRNA was similar to binding to BLT dsRNA without the cyclic phosphate block (*Figure 3—figure supplement 2C D, and H*, and 2 H), and $K_d$ values were nearly identical (*Supplementary file 1a*). This is consistent with the idea that in the presence of ATP, binding is predominantly to the helicase domain of DCR-1 or DRH-1, similar to what is observed in dmDcr2 (*Singh et al., 2021*). In the absence of ATP, DCR-1•DRH-1•RDE-4 binds both types of BLT dsRNA with similar affinities (*Figure 3—figure supplement 2E*, compare *Table 1* and *Supplementary file 1a*), but DCR-1•DRH-1 binds 2.5-fold better to BLT dsRNA than it does to cyclic blocked BLT dsRNA (*Figure 3—figure supplement 2F and H*; compare *Table 1* and *Supplementary file 1a*). This suggests that the 2', 3'-cyclic phosphate blocks terminus binding to the PAZ/Platform domain but the presence of RDE-4 in the wildtype complex mutes this effect.

## Cryo-EM analyses give insight into ATP-independent and ATP-dependent cleavage mechanisms

To better understand how each protein in the *C. elegans* antiviral complex contributes to dsRNA processing, we used cryo-EM to determine structures of DCR-1•DRH-1•RDE-4 in the presence of dsRNA substrates, with or without nucleotide. We obtained a 6.1 Å reconstruction of DCR-1•DRH-1•RDE-4 bound to a 52-BLT dsRNA (*Figure 4A*, *Figure 4—figure supplement 1*). Our density map revealed the canonical 'L shape' of Dicer (*Lau et al., 2012*; *Lau et al., 2009*; *Liu et al., 2018*; *Sinha et al., 2018*; *Yamaguchi et al., 2022*), which enabled rigid-body fitting of a human Dicer model. The reconstruction contained extra density at the Hel2i subdomain of the DCR-1 helicase domain, a known binding site of dsRBPs, such as human TRBP and *Drosophila* R2D2 (*Liu et al., 2018*; *Yamaguchi et al., 2022*). Indeed, when fitting the human Dicer•TRBP model (*Liu et al., 2018*) into our density, the extra density was accommodated by the third dsRNA binding motif (dsRBM) of TRBP, and by analogy, we propose this density corresponds to a single dsRBM of RDE-4 (*Figure 4A*). In our 3D reconstructions, we also observed additional density interacting with Dicer's helicase domain near the Hel1 subdomain. In this density, we were able to fit AlphaFold2 models of DRH-1 NTD and helicase domains as separate rigid bodies (*Figure 4A*; *Jumper et al., 2021*; *Varadi et al., 2022*). These fits showed the NTD of DRH-1 directly interacting with DCR-1's helicase domain, with DRH-1's helicase and CTD domains extended away from DCR-1 and bound to the end of the 52-BLT dsRNA (*Figure 4A*).

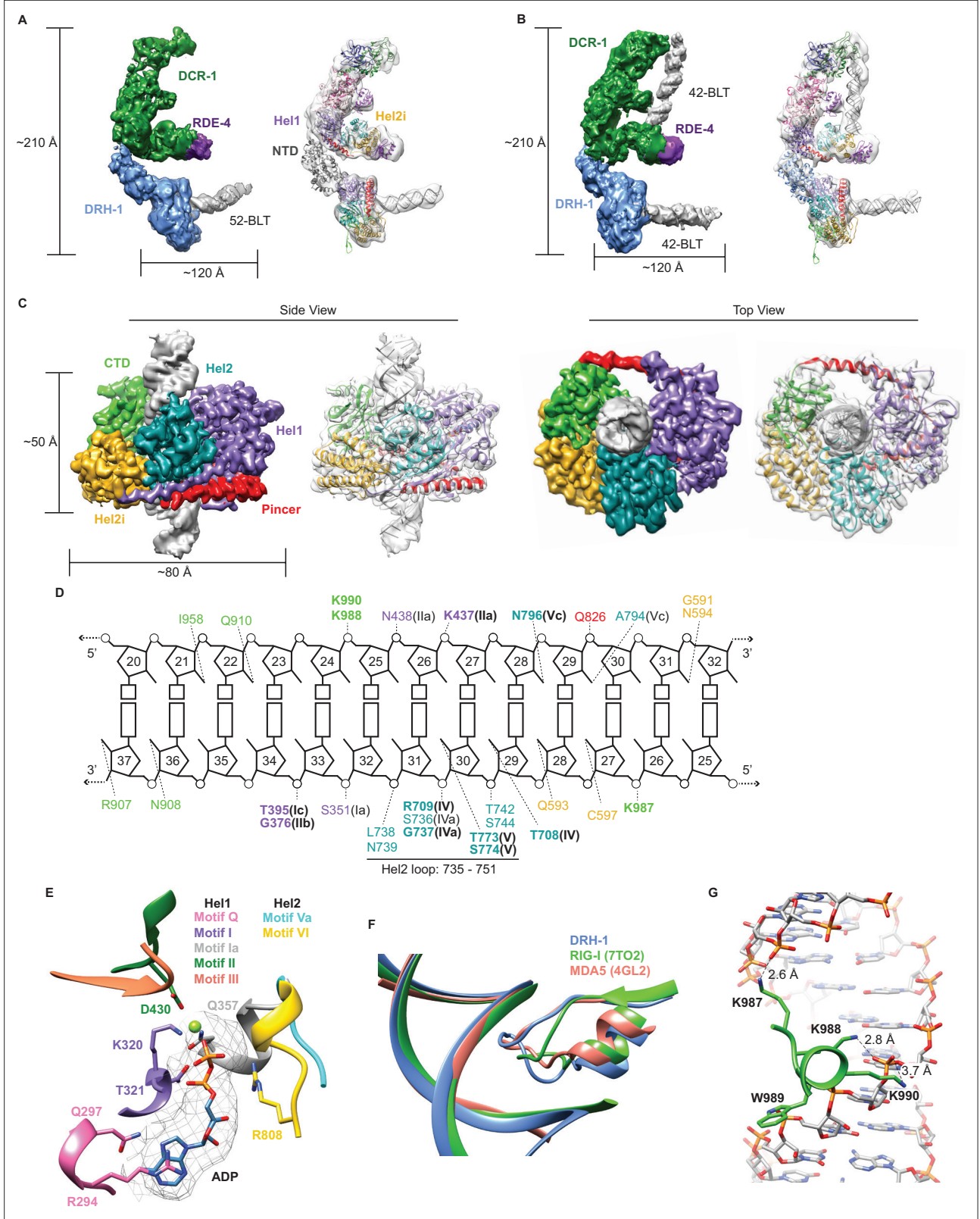

**Figure 4.** Cryo-EM analyses provide insight into ATP-dependent and ATP-independent cleavage mechanisms. (**A**) 6.1 Å reconstruction of DCR-1•DRH-1•RDE-4. DCR-1 and RDE-4 densities were fitted with human Dicer in complex with TRBP (PBD 5ZAK) (*Liu et al., 2018*), DRH-1 density was fitted with an AlphaFold2 model of DRH-1, and a 52-BLT A-form dsRNA was built in Chimera. Color of protein names correlates with model colors. The domains of DCR-1, DRH-1, and RDE-4 are color coded the same as in *Figure 1A*. For simplicity, only domains discussed in the text are labeled. See also *Figure 4—*

*Figure 4 continued on next page*

*Figure 4 continued*

*figure supplement 1*. (**B**) 7.6 Å reconstruction of DCR-1•DRH-1•RDE-4. DCR-1 and RDE-4 densities were fit with PBD 5ZAL (*Liu et al., 2018*), DRH-1 density was fit with the AlphaFold2 prediction of DRH-1, and a 42-BLT A-form dsRNA was built in Chimera. See also *Figure 4—figure supplement 2*. (**C**) 3D reconstruction at 2.9 Å fitted with a refined model of the helicase and CTD domains of DRH-1 and dsRNA. See also *Figure 4—figure supplement 3* and *Supplementary file 1b*. (**D**) The interactions between DRH-1 and dsRNA, determined with a 3.7 Å cutoff for contacts. Bold residues indicate residues that are identical with RIG-I, MDA5, or both. See also *Figure 4—figure supplement 4*. (**E**) ADP, Mg$^{++}$, and surrounding helicase motifs. Colors: motif Q (pink), motif I (purple), motif Ia (gray), motif II (green), motif III (coral), motif Va (cyan), and motif VI (yellow). The density of ADP is shown in mesh. (**F**) Conserved loop in the Hel2 domain is inserted into the dsRNA major groove. DRH-1=blue, residues 733–756; RIG-I (7TO2) (*Wang and Pyle, 2022*)=green, residues 659–679; MDA5 (4GL2) (*Wu et al., 2013*)=salmon, residues 752–773. Note that this figure includes residues on each side of the loop. See also *Figure 4—figure supplement 4*. (**G**) Lysines 987, 988, and 990 interact with the dsRNA backbone. See also *Figure 4—figure supplement 4*.

The online version of this article includes the following figure supplement(s) for figure 4:

**Figure supplement 1.** Validation of DCR-1•DRH-1•RDE-4 structure with 52 BLT dsRNA, no ATP.

**Figure supplement 2.** Validation of DCR-1•DRH-1•RDE-4 structure with 42 BLT dsRNA.

**Figure supplement 3.** Validation of DRH-1 structure with 106 3'ovr dsRNA.

**Figure supplement 4.** Sequence alignment of RLRs and Dicer helicase and DUF domains.

**Figure supplement 5.** Comparison of DRH-1 with RIG-I and MDA5.

In a separate reconstruction, we solved DCR-1•DRH-1•RDE-4 bound to 42-BLT dsRNA to 7.6 Å (*Figure 4B*, *Figure 4—figure supplement 2*). This structure was similar to our 6.1 Å reconstruction, with the exception of additional dsRNA density bound to DCR-1 (compare *Figure 4A and B*). We fit the human Dicer•TRBP•pre-miRNA Class 1 structure (*Liu et al., 2018*) as a rigid body into our complex and found that there is a slight difference in the position of the dsRNA; however, like the human Dicer structure, our density indicates that the 42-BLT dsRNA is positioned away from the RNase III active sites. This state likely represents a pre-dicing conformation, similar to many other studies that show the importance of the PAZ domain in positioning pre-miRNA for ATP independent cleavage (*Liu et al., 2018*; *Zhang et al., 2004*; *Zhang et al., 2002*). In both of our reconstructions, we observed DRH-1 bound to dsRNA termini (*Figure 4A and B*); however, it was still unclear how DRH-1 utilizes ATP to promote ATP-dependent cleavage. We speculated that the dsRNA bound to DRH-1 was trapped in an initial dsRNA binding state, because our reconstructions thus far excluded ATP and were performed in the absence of nucleotide or with ATPγS.

To understand how ATP hydrolysis was coupled to cleavage, we prepared cryo-EM specimens of DCR-1•DRH-1•RDE-4 in the presence of ATP, and added Ca$^{2+}$ in addition to Mg$^{2+}$ to inhibit dsRNA cleavage (see Materials and methods). Under these conditions, we obtained a 2.9 Å reconstruction of the helicase and CTD domains of DRH-1 bound to the middle region of the dsRNA, rather than its terminus (*Figure 4C*, *Figure 4—figure supplement 3*), suggesting that DRH-1 hydrolyzes ATP to translocate along dsRNA. Density for DCR-1, RDE-4, and the NTD of DRH-1 was not observed in this reconstruction, presumably due to increased flexibility in the presence of ATP.

The reconstruction showed DRH-1 in a closed conformation where Hel1, Hel2, Hel2i, and the CTD were wrapped around the dsRNA to make a network of contacts with both strands (*Figure 4C and D*). The N-terminal RecA fold (Hel1) was positioned next to the second RecA fold (Hel2). The ATP-binding pocket at the interface of these domains contained density for ADP and Mg$^{2+}$, stabilized by contacts from well conserved helicase motifs including Q, I, Ia, II, III, Va, and VI (*Figure 4E*, *Figure 4—figure supplement 4*). In the Hel1 subdomain, direct binding to ADP-Mg$^{2+}$ involves residues from motif Q (Q297), motif I (K320 and T321), motif Ia (Q357), and motif II (D430) (*Figure 4E*). Motif Q recognizes adenine and mediates binding specificity to ATP as in other SF1 and SF2 helicases with R294's sidechain forming stacking interactions with the adenine base, in addition to the hydrogen bonds between Q297 and adenine N$_6$ and N$_7$ (*Figure 4E*; *Civril et al., 2011*; *Cordin et al., 2004*; *Fairman-Williams et al., 2010*). In the Hel2 subdomain, R808 from Motif VI is positioned to make contact with the beta phosphate of the ADP molecule. In Hel2, we also found a conserved loop between motifs IVa and IVb was inserted into the major groove of the dsRNA (*Figure 4F*, *Figure 4—figure supplement 4*), with a concomitant widening of the groove (*Figure 4F*, *Figure 4—figure supplement 5A and B*). The analogous loop of MDA5 interacts with a widened major groove and is crucial for dsRNA-dependent ATP hydrolysis by MDA5 (*Wu et al., 2013*; *Yu et al., 2018*). In RIG-I, motif IVa is disordered to accommodate a 5'ppp or m7G moiety, but is observed with dsRNA containing a 5'OH, or when

RIG-I is internally bound to dsRNA (*Devarkar et al., 2016*; *Wang and Pyle, 2022*). Deletion of this loop in RIG-I reduces ATP hydrolysis and inactivates signaling (*Devarkar et al., 2016*). Furthermore, RIG-I's loop contains two threonine residues (T667 and T671) that when phosphorylated impair RIG-I translocation and signaling (*Devarkar et al., 2018*; *Willemsen et al., 2017*, *Figure 4—figure supplement 4*, see arrows pointing down). Interestingly, DRH-1's Hel2 loop has two serines (S741 and S745) that align with RIG-I's threonines (*Figure 4—figure supplement 4*), raising the possibility that DRH-1 might also be regulated by phosphorylation state.

The Hel2i subdomain, an alpha helical insertion domain unique to RLRs and metazoan Dicers, is comprised of five alpha helices and is inserted between Hel2 and the CTD. The pincer subdomain forms the canonical V-shape that acts as a bridge connecting Hel1, Hel2, and the CTD. In the CTD, we observed a patch of lysine residues (K987, K988, and K990) that make extensive contact with phosphates in the RNA backbone (*Figure 4D and G*). Interestingly, *Guo et al., 2013* demonstrated that a *C. elegans drh-1* null mutant animal carrying an extrachromosomal array encoding DRH-1 with point mutations K988A, W987A, and K990A, failed to rescue RNAi, which suggests these residues are key in maintaining contact with dsRNA. K988 and W989 are conserved in RLRs, indicating these are important residues in the CTD (*Figure 4—figure supplement 4*). We also found that the CTD contains a zinc structural ion that is conserved with RLRs (*Figure 4—figure supplement 4*, see arrows pointing up, and figure supplement 5C). Overall, the tertiary structure of DRH-1 was well conserved with RLRs (*Figure 4—figure supplement 5D*). The RMSD of DRH-1 to an internally bound RIG-I (PDB 7TO2 *Wang and Pyle, 2022*) was 1.706 Å, while to MDA5 (PDB 4GL2 *Wu et al., 2013*) it was 1.708 Å.

## The NTD of DRH-1 is autoinhibitory

When full-length DRH-1 was purified by itself, we found that it could not hydrolyze ATP (*Figure 5A*, DRH-1). Since our structural studies suggested that DRH-1 translocated along dsRNA, we considered the possibility that interaction of its NTD with DCR-1's helicase domain released DRH-1 from an autoinhibited state. This was an attractive model by analogy to RIG-I, which contains two N-terminal caspase activation and recruitment domains (CARDs) (*Figure 1A*) involved in autoinhibition. In the absence of dsRNA, the second CARD (CARD2) interacts with RIG-I's Hel2i domain to stabilize an autoinhibited conformation and prevent signaling of an interferon response (*Kowalinski et al., 2011*; *Ramanathan et al., 2016*). In the presence of its optimal substrate, a 5'ppp BLT dsRNA, RIG-I's CTD and helicase domains bind the dsRNA terminus, promoting a conformational change that disrupts the CARD2 interaction and autoinhibition. This allows RIG-I to translocate along dsRNA, form oligomers, and signal an interferon response (*Devarkar et al., 2018*; *Peisley et al., 2013*). In the absence of DCR-1 and RDE-4, full-length DRH-1 cannot hydrolyze ATP with dsRNA (*Figure 5A*). However, we found that deletion of its NTD (ΔNTD DRH-1), allowed ATP hydrolysis in a terminus-dependent manner (*Figure 5A and B*). Based on these results and our structural data (*Figure 4A and B*), we propose that the NTD of DRH-1, like RIG-I's CARD domains, interacts with its own helicase domain to promote an autoinhibited state. While binding to dsRNA relieves autoinhibition for RIG-I, for DRH-I we propose it is the binding of its NTD to DCR-1's helicase domain that relieves DRH-1's autoinhibited state (*Figure 4*). Moreover, ΔNTD DRH-1 hydrolyzes ATP more efficiently in the presence of blunt dsRNA as opposed to 3'ovr dsRNA, suggesting that, like RIG-I, DRH-1 recognizes dsRNA termini (*Figure 5A and B*).

## DCR-1•DRH-1•RDE-4 processively cleaves BLT dsRNA

Given that DCR-1•DRH-1•RDE-4 cleaves BLT dsRNA in an ATP-dependent manner (*Figure 1C–E*), and our data suggest DRH-1 hydrolyzes ATP for translocation (*Figures 2H and 4C*), we wondered if translocation was coupled to processive cleavage. To test this, we performed trap experiments where DCR-1•DRH-1•RDE-4 was allowed to cleave [32]P-internally labeled BLT dsRNA for a short time, followed by addition of excess nonradiolabeled (cold) dsRNA. When cold dsRNA was added second, we saw a small continued accumulation of siRNA over time, albeit the increase was slight (*Figure 5C and D*), consistent with the idea that DCR-1•DRH-1•RDE-4 remained associated with dsRNA during successive cleavage events. Additional support for the role of DRH-1 in processive cleavage came from single turnover cleavage assays using [32]P-internally labeled BLT dsRNA to allow visualization of the initial cleavage event from the terminus (e.g. *Figure 1*), as well as siRNAs from internal cleavage (*Figure 5E*). For WT complex in the presence of ATP, the 26nt and 27nt siRNA bands that correspond

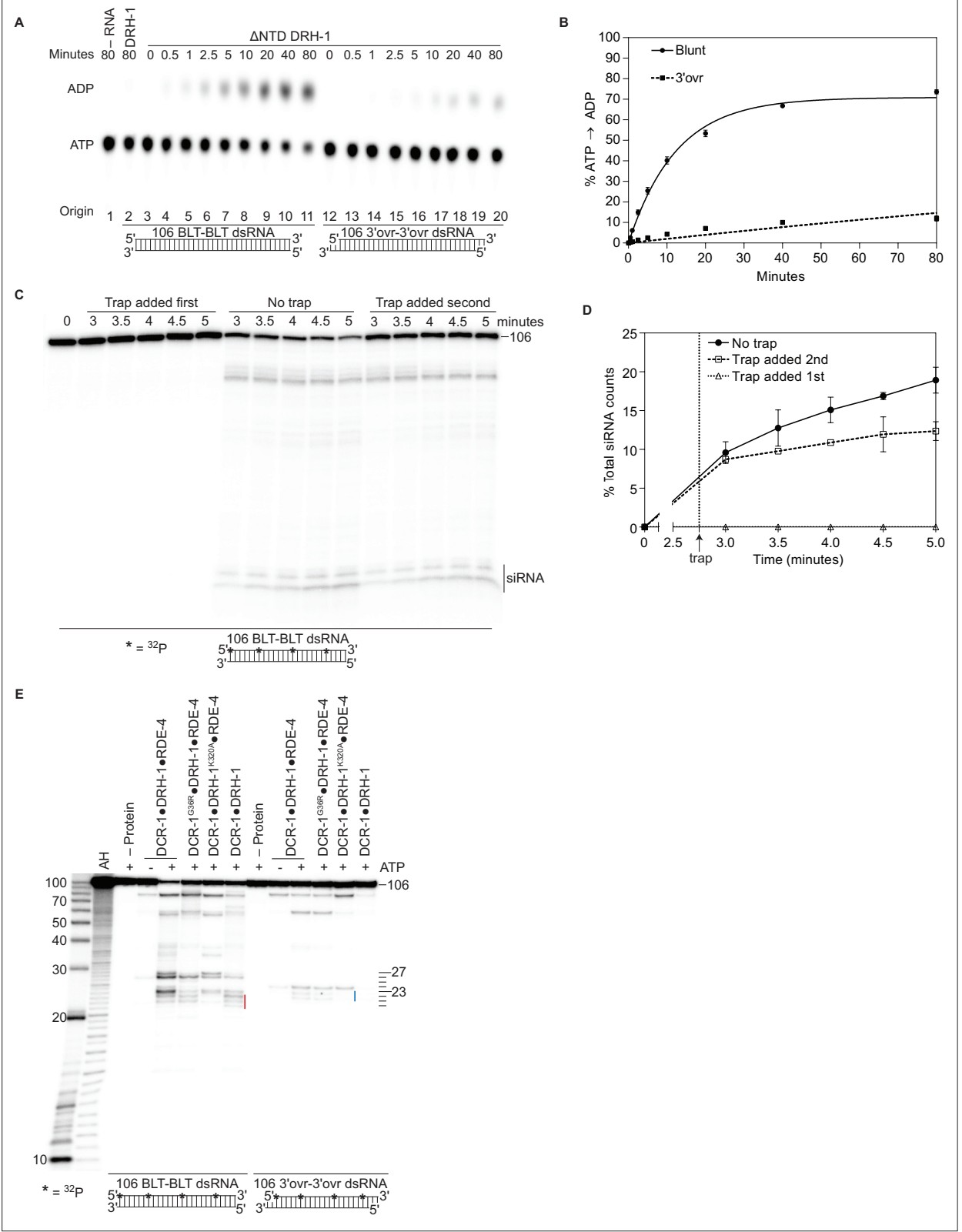

**Figure 5.** Autoinhibition of DRH-1 is relieved to promote processive cleavage. (**A**) Full length DRH-1 (25 nM) was incubated with 106 BLT dsRNA (400 nM) with 100 μM α-³²P-ATP at 20 °C (lane 2). ΔNTD DRH-1 (25 nM) was incubated without RNA (-RNA; lane 1) or with 400 nM 106 BLT (lanes 3–11) or 3'ovr dsRNA (lanes 12–20) with 100 μM α-³²P-ATP at 20 °C. ATP hydrolysis was monitored by TLC, and a representative PhosphorImage is shown. Positions of origin, ATP, and ADP are indicated. (**B**) Quantification of ATP hydrolysis assay as in A. Data points are mean ± SD (n=3) with symbol key

*Figure 5 continued on next page*

*Figure 5 continued*

in graph. (**C and D**) A representative PhosphorImage (**C**) shows trap experiments using 50 nM DCR-1•DRH-1•RDE-4 and 1 nM $^{32}$P-internally labeled 106 BLT dsRNA ±2000 nM 52 BLT cold dsRNA. Quantification (**D**) shows mean ± SD (n=3). Cold dsRNA was added at 2.75 minutes as indicated by the vertical dotted line and arrow. (**E**) Single-turnover cleavage assays of 106 BLT or 106 3'ovr dsRNA (1 nM) with indicated protein complex (25 nM)±5 mM ATP as indicated. Sense strand was $^{32}$P-internally labeled to allow visualization of intermediates, with red line noting internal cleavage products of 21–23 nts and blue line noting 22–23 nt products. Products were separated by 17% denaturing PAGE, and a representative PhosphorImage is shown (n=3). Left, marker nucleotide lengths. AH, alkaline hydrolysis.

The online version of this article includes the following source data for figure 5:

**Source data 1.** Raw digital image of thin-layer chromatography plate used in *Figure 5A*.

**Source data 2.** Raw digital image of cleavage phosphorimager plate used in *Figure 5C*.

**Source data 3.** Raw digital image of cleavage phosphorimager plate used in *Figure 5E*.

to the initial cleavage from the terminus were observed, as well as a cluster of smaller siRNAs (~21 – 23nts). Based on prior studies (*Welker et al., 2011*), the 23nt species is predicted to be comprised of siRNAs produced from the initial cleavage from the opposite end, as well as internal cleavage, while the 21nt and 22nt species derive exclusively from internal cleavage (*Figure 5E*, see red line). Importantly, the DCR-1•DRH-1$^{K320A}$•RDE-4 complex showed the greatest loss in the production of internal siRNAs (*Figure 5E*). A similar loss of internal siRNA was observed in the presence of $^{32}$P-internally labeled 3'ovr dsRNA only when DRH-1's helicase is mutated (*Figure 5E*, blue line 22 – 23nts). Since DRH-1 is required for ATP hydrolysis (*Figure 2F and H*), it is possible that DRH-1 is needed to allow DCR-1 to processively cleave dsRNA. This observation is supported by Coffman *et. al*, who showed that *C. elegans* strains containing *drh-1* mutations accumulated siRNAs corresponding to terminal regions of the Orsay virus genome, but lost viral siRNAs mapping to internal regions (*Coffman et al., 2017*).

## Discussion

Here, we report the first biochemical characterization of *C. elegans* Dicer in a recombinant form. Key to the successful purification of DCR-1 was its co-expression and purification with two proteins known to associate with DCR-1: the RIG-I ortholog DRH-1, and the dsRBP RDE-4 (*Tabara et al., 2002*; *Thivierge et al., 2012*). Prior studies show that all three proteins are required to cleave viral dsRNA into viral siRNAs during the *C. elegans* antiviral response (*Ashe et al., 2013*; *Guo et al., 2013*; *Lu et al., 2009*; *Tabara et al., 2002*; *Thivierge et al., 2012*). Our structural studies reveal physical interactions between the three proteins, and combined with our biochemical assays, provide insight into the mechanism of a molecular motor that requires two helicases and a dsRBP. By comparing the wildtype antiviral complex to complexes with a Walker A mutation in the helicase domain of DCR-1 or DRH-1, or a complex lacking RDE-4, we gained insight into the contributions of the three proteins to the processing of dsRNA.

### Relationship to previously characterized Dicer activities

Vertebrates primarily rely on RLRs and the interferon response for antiviral defense, and their Dicer enzymes have not been observed to require ATP. By contrast, invertebrate antiviral defense occurs through the RNAi pathway using Dicer enzymes that are ATP dependent. The most biochemically well characterized invertebrate antiviral Dicer is dmDcr2, which cleaves dsRNA in an ATP- and dsRNA terminus-dependent manner in vitro, without accessory factors; studies in flies indicate these same properties exist in vivo (*Donelick et al., 2020*). We find that the *C. elegans* antiviral complex is also ATP and terminus dependent. Like dmDcr2, the *C. elegans* antiviral complex cleaves BLT dsRNA more efficiently than 3'ovr dsRNA (*Figure 1E*), and with 106-dsRNA (1 nM) the single turnover cleavage rates for purified dmDcr2 (30 nM) with ATP (BLT, $k_{obs}$, 0.19 min$^{-1}$; 3'ovr, $k_{obs}$, 0.01 min$^{-1}$; *Sinha et al., 2015*) are almost identical to the rates observed with the *C. elegans* complex (25 nM) (BLT, $k_{obs}$, 0.14 min$^{-1}$, 3'ovr, $k_{obs}$, 0.006 min$^{-1}$, *Table 1*). We find it remarkable that the enzymatic properties for dsRNA cleavage are so similar, given that dmDcr2 orchestrates cleavage with a single protein, while two helicases and a dsRNA binding protein cooperate in the *C. elegans* reaction.

There are also differences between the two antiviral activities. Like dmDcr2, for the *C. elegans* antiviral complex, ATP hydrolysis is faster with BLT dsRNA ($k_{obs}$ = 0.11 min$^{-1}$) compared to 3'ovr dsRNA

(0.002 min-1; *Table 1*). However, a mutation in the Walker A motif of DCR-1 reduces both of these values, but a similar mutation in DRH-1 completely eliminates detectable hydrolysis, indicating DRH-1 is the primary driver of ATP hydrolysis in the *C. elegans* antiviral complex. Indeed, DRH-1 alone hydrolyzes ATP with BLT dsRNA at almost the same rate as the complex ($k_{obs}$, 0.08 min$^{-1}$) in a terminus-dependent manner (*Figure 5A*), emphasizing the key role it plays in the antiviral complex. That said, we observed a one nucleotide longer siRNA that was dependent on ATP hydrolysis by DCR-1 (*Figure 2A*), indicating that DCR-1 contributes to the ATP-dependence of the reaction. Also, although we see ATP-dependent cleavage, the cleavage pattern is remarkably different than dmDcr2. In the presence of ATP, dmDcr2 produces heterogeneous sized siRNA products, whereas the *C. elegans* antiviral complex produces discrete siRNA bands. These discrete siRNA bands are more similar to human Dicer, although human Dicer does not cleave dsRNA in an ATP-dependent manner (*Park et al., 2011*; *Zhang et al., 2002*).

Another difference is that, while dmDcr2 readily cleaves 3'ovr dsRNA in the absence of ATP, it has a strict requirement for ATP in processing BLT dsRNA. By contrast we observe ATP independent cleavage for both 3'ovr and BLT dsRNA with the *C. elegans* complex, and our studies indicate RDE-4 is important for this ATP-independent cleavage. In complexes lacking RDE-4, cleavage is only detected in the presence of ATP and BLT dsRNA, and even this reaction is barely detectable (*Figure 2A–C*). Our biochemical studies indicate that RDE-4 is important for dsRNA binding, and this is not surprising since it consists of three dsRBMs. The DCR-1•DRH-1 complex, which lacks RDE-4, shows approximately 10-fold weaker binding for BLT dsRNA (5.18 nM) and 32-fold weaker binding (28.64 nM) for 3'ovr dsRNA, suggesting that RDE-4 dampens the difference in affinity for different termini conferred by DCR-1 and/or DRH-1 (*Table 1*). In this regard, RDE-4 is similar to the dsRBP Loquacious-PD (Loqs-PD), an accessory factor of dmDcr2 that is required for processing endogenous, but not viral, dsRNA. Loqs-PD decreases dmDcr2's terminus dependence, and for example, while BLT dsRNA is cleaved 19-fold faster than 3'ovr dsRNA in the absence of Loqs-PD, addition of this accessory factor decreases the rate difference to about twofold.

## A model for cleavage by the *C. elegans* antiviral complex

As illustrated in *Figure 6*, a key feature of the antiviral complex is the interaction of DRH-1's NTD with DCR-1's helicase domain. When assaying DRH-1 alone, we observed ATP hydrolysis only when the NTD was removed (*Figure 5A*), and we speculate this autoinhibition is relieved by interaction with DCR-1's helicase domain. While we only observe density for a single dsRBM in each of our structures, in the models of *Figure 6* we show all three motifs of RDE-4, consistent with other models of Dicer and dsRBPs (*Liu et al., 2018*; *Yamaguchi et al., 2022*). By analogy to roles of dsRBMs in RNase III enzymes, it is possible that one or more of RDE-4's dsRBMs stabilizes dsRNA in a bent conformation that facilitates catalysis (*Hansen et al., 2019*). Hence, in our model, RDE-4 positions dsRNA in the RNase III active sites for ATP-independent and -dependent mechanisms (illustration 4 in *Figure 6A*); however, future studies will be required to validate this.

In the absence of nucleotide or in the presence of ATPγS, our cryo-EM structures show DRH-1 binding to the end of dsRNA, while addition of ATP shows DRH-1 localized to internal regions, consistent with the idea that DRH-1 translocates along dsRNA. We also observed that DRH-1 hydrolyzes ATP in a terminus-dependent manner (*Figure 5A and B*), and thus, we show DRH-1 as the entry point for dsRNA (illustration 1 in *Figure 6A*), and by analogy to RLRs, propose that DRH-1 uses the energy of ATP hydrolysis to discriminate BLT and 3'ovr dsRNA, or nonself versus self (*Devarkar et al., 2018*). As yet, how dsRNA proceeds from interacting exclusively with DRH-1 to then associating with DCR-1 for cleavage is unclear. As depicted, in our favorite model there is a conformational change that is more probable after recognition of a BLT terminus by DRH-1, that juxtaposes the dsRNA-bound helicase domain of DRH-1 with DCR-1's helicase domain for subsequent threading to the RNase III active site (illustrations 2 and 3 in *Figure 6A*). While our studies indicate DRH-1 plays the major role in ATP hydrolysis, our studies of a Walker A mutation in DCR-1 indicate its helicase domain also contributes to ATP hydrolysis and is required to produce an ATP-dependent 27nt when processing BLT dsRNA (*Figure 2A*). This suggests DCR-1 is involved in recognizing the BLT terminus, and thus we show dsRNA interacting with DCR-1's helicase domain. We show dsRNA threading processively through the complex, as suggested by our trap experiments, as well as experiments showing that a mutation in DRH-1's helicase domain precludes generation of internal siRNAs (*Figure 5C–E*). Since the K320A

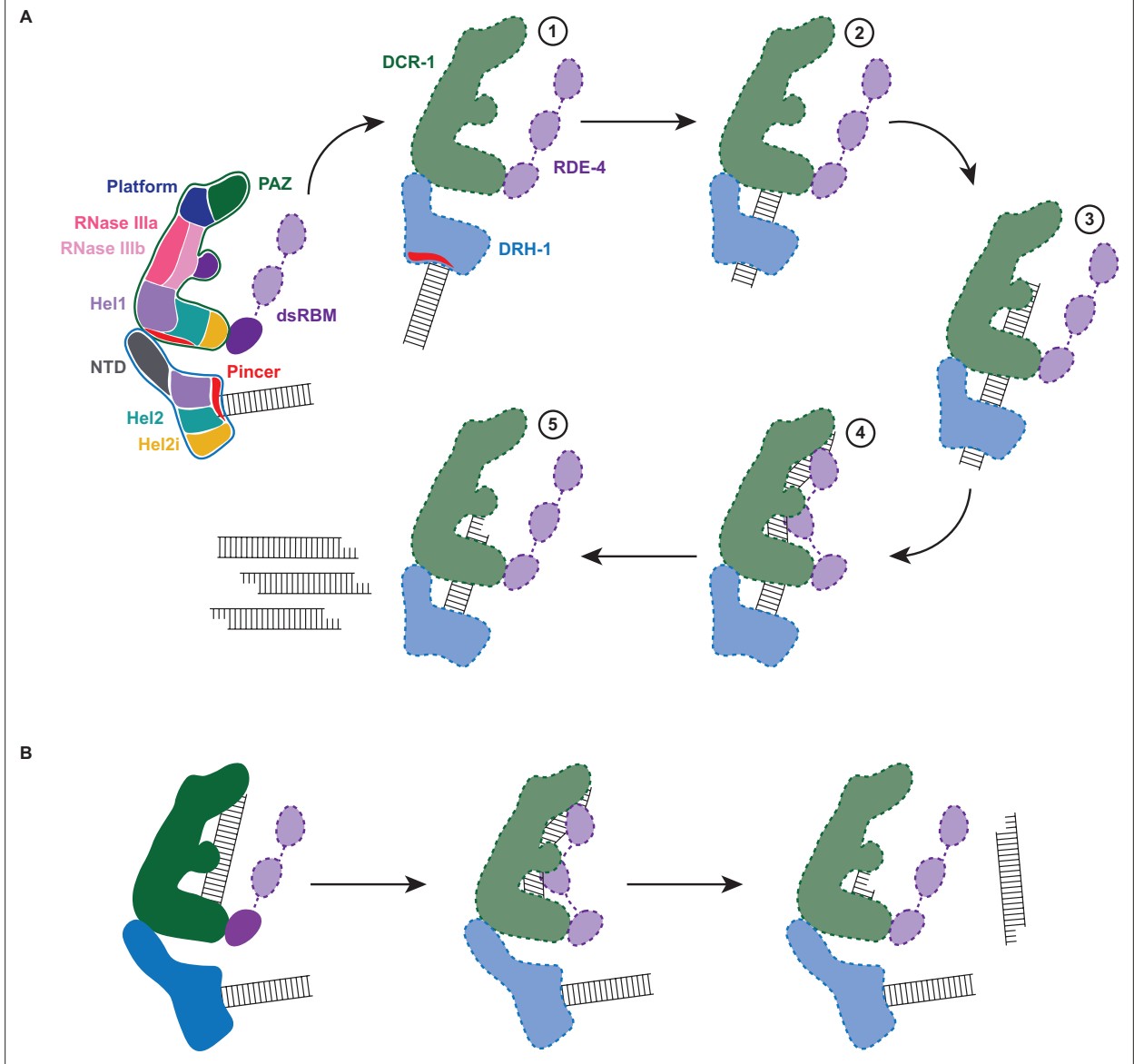

**Figure 6.** Working model of dsRNA cleavage by DCR-1•DRH-1•RDE-4. (**A**) Model for ATP-dependent cleavage. Far left shows cartoon of complex with domains color-coded. Illustrations 1–5 show proposed steps as discussed in text, with DCR-1 (green), DRH-1, (blue), and RDE-4 (purple). Solid lines indicate structures observed in our cryo-EM studies, while dashed lines indicate proposed intermediates not yet observed. Pincer subdomain (red) is colored to orient proposed DRH-1 conformational change. Additional details in text. (**B**) Model for ATP-independent cleavage, with colors and solid and dashed lines as in A.

mutation in DRH-1's helicase domain precludes ATP hydrolysis (**Figure 2F**), the cleavage observed from the [32]P-labeled dsRNA terminus either occurs without ATP hydrolysis (**Figure 2A**) or relies solely on hydrolysis from DCR-1. While the complex containing DRH-1[K320A] does not show detectable hydrolysis, we cannot exclude the possibility that low levels of hydrolysis by DCR-1 are below the sensitivity of our assay. These observations suggest that production of the initial siRNA from a dsRNA end does not require DRH-1's ATP hydrolysis activity, but continued threading does.

In our 7.6 Å structure acquired in the absence of ATP or with ATPγS, we observed one dsRNA interacting with DCR-1, and one with DRH-1, and speculate dsRNA associated with DCR-1 is poised for ATP-independent cleavage mediated by the Platform•PAZ domain (**Figure 6B**). We note however, that this structure would also be consistent with a mechanism whereby, after interaction with DRH-1, the dsRNA is handed off to DCR-1 without threading.

## Why does *C. elegans* Dicer require accessory factors?

While it seems likely that Dicer was dedicated to antiviral defense in the common ancestor of animals (*Aderounmu et al., 2023*; *Mukherjee et al., 2013*), modern day animal Dicers have multiple roles, including processing dsRNA to produce mature miRNAs and endogenous siRNAs, and for invertebrates, processing viral dsRNA to produce viral siRNAs. In Arthropods, gene duplication led to Dicer enzymes specialized for different functions. *Drosophila melanogaster* uses dmDcr-1 for miRNA processing and dmDcr2 for processing viral dsRNA in the antiviral response. There is a need for specialization since recognition of a pre-miRNA, and the single round of distributive cleavage that generates a mature miRNA (*Lee et al., 2023*; *Park et al., 2011*), places different demands on Dicer than the recognition of a nonself viral dsRNA and the processive cleavage that most efficiently processes viral dsRNA to siRNA.

*H. sapiens* and *C. elegans* have only a single Dicer, and it is interesting to compare how these organisms have delegated requisite functions. To deal with antiviral defense, vertebrates, including humans, use RLRs to recognize viral dsRNA and trigger an interferon response. Presumably, this allowed human Dicer to specialize for miRNA processing. *C. elegans* does not have an interferon pathway, and its single Dicer enzyme is responsible for processing miRNAs, producing endogenous siRNAs, and promoting a robust antiviral response. An attractive idea is that accessory factors are key to allowing DCR-1 to function in these multiple pathways and to discriminate between different dsRNA substrates. Interestingly, *C. elegans* Dicer interacts with distinct RLR orthologs for its different functions. Our study focused on characterizing the antiviral complex, but another RLR homolog, DRH-3, associates with *C. elegans* DCR-1 to produce endogenous siRNA (*Thivierge et al., 2012*). DRH-3 has been biochemically characterized and, like DRH-1, has robust ATP hydrolysis activity (*Matranga and Pyle, 2010*).

# Materials and methods

**Key resources table**

| Reagent type (species) or resource | Designation | Source or reference | Identifiers | Additional information |
|---|---|---|---|---|
| Antibody | Anti-drh-1 (N-term) antibody | Sigma-Aldrich | Cat# SAB1300243 | |
| Antibody | DYKDDDDK (Flag) Tag Polyclonal Antibody | ThermoFisher | Cat# PA1-984B | |
| Antibody | dcr-1 polyclonal antibody | Fischer Scientific | Cat# PAB3792 | |
| Antibody | Anti-rde-4 | Thomas Duchaine | N/A | |
| Cell line | Sf9 (*Spodoptera frugiperda*) cells | Expression Systems | Cat# 94–001 S | |
| Commercial kit | Baculovirus titering kit | Expression Systems | Cat# 97 | |
| Commercial kit | biGBac kit | Addgene | Cat# 1000000088 | Rapid generation of baculoviral expression constructs |
| Recombinant DNA reagent | DCR-1 in plib vector | This paper | N/A | plib vector is from biGBac kit |
| Recombinant DNA reagent | DCR-1$^{G36R}$ in plib vector | This paper | N/A | |
| Recombinant DNA reagent | OSF-DRH-1 in plib vector | This paper | N/A | DRH-1 was modified in house to add OSF tag |
| Recombinant DNA reagent | OSF-ΔNTD-DRH-1 in plib vector | This paper | N/A | |
| Recombinant DNA reagent | OSF-DRH-1$^{K320A}$ in plib vector | This paper | N/A | |
| Recombinant DNA reagent | RDE-4 in plib vector | This paper | N/A | |
| Recombinant DNA reagent | DCR-1•OSF-DRH-1•RDE-4 in pbig vector | This paper | N/A | pbig vector is from biGBac kit |
| Recombinant DNA reagent | DCR-1$^{G36R}$•OSF-DRH-1•RDE-4 in pbig vector | This paper | N/A | |

*Continued on next page*

*Continued*

| Reagent type (species) or resource | Designation | Source or reference | Identifiers | Additional information |
|---|---|---|---|---|
| Recombinant DNA reagent | DCR-1•OSF-DRH-1$^{K320A}$•RDE-4 in pbig vector | This paper | N/A | |
| Recombinant DNA reagent | DCR-1•OSF-DRH-1 in pbig vector | This paper | N/A | |

## Plasmids and cloning

DCR-1•DRH-1•RDE-4, DCR-1$^{G36R}$•DRH-1•RDE-4, DCR-1•DRH-1$^{K320A}$•RDE-4, and DCR-1•DRH-1 constructs were cloned into one vector as described (*Weissmann et al., 2016*). Briefly, for DCR-1•DRH-1•RDE-4, the coding sequences of *dcr-1*, *drh-1*, and *rde-4* were separately amplified using primers that contained overhangs to allow insertion of each gene between the BamHI and HIndIII sites of the plib vector to create a gene expression cassette (GEC) consisting of a polyhedrin promoter (polh), cDNA of gene, and SV40-terminator (term). The circularized products were transformed into DH5α chemically competent cells. Individual colonies were isolated, plasmids purified, and sent to Genewiz to confirm sequence of each GEC. The *dcr-1* GEC was amplified with cas III forward and cas V reverse primers, the *drh-1* GEC was amplified with cas I forward and cas I reverse primers, and the *rde-4* GEC was amplified with cas II forward and cas II reverse primers. The linearized pBig1b vector and PCR products were recombined in a Gibson assembly reaction using NEBuilder HiFi DNA Assembly Master Mix to create a circular product containing a polygene cassette (PGC). The circular product was transformed into MegaX DH10B T1 electrocompetent cells. Plasmids were purified from individual colonies and the SwaI and PmeI restriction sites were used to analyze the presence of all three genes. Plasmids that showed the correct digest patterns were sent to Genewiz to confirm the presence and sequence of each gene. Plib constructs were used as templates for site-directed mutagenesis to introduce point mutations in the Walker A sequences to create DCR-1$^{G36R}$ and DRH-1$^{K320A}$. We then followed the necessary steps to insert the indicated coding sequences into the pBig1b vector to create the other PGCs: DCR-1$^{G36R}$•DRH-1•RDE-4, DCR-1•DRH-1$^{K320A}$•RDE-4, and DCR-1•DRH-1.

## Expression system

Protein complexes were expressed in *Spodoptera frugiperda* (Sf9) cells using the modified Bac-to-Bac Baculovirus Expression System as described (*Sinha and Bass, 2017*).

## Protein purification

Protein complexes were purified using a modified method of *Sinha and Bass, 2017* to optimize for *C. elegans* proteins. Cell pellets were resuspended in 5.5 times the pellet volume in buffer (25 mM HEPES [pH 7.5]; 10 mM KOAc; 2 mM Mg(OAc)$_2$; 100 mM KCl; 1 mM TCEP; 10% glycerol) supplemented with 1% Triton X-100, 20 nM Avidin, 250 mg/ml DNase I Grade II, cOmplete EDTA-free protease inhibitor (1 tablet per 25 ml buffer), 0.7% (vol/vol) Protease Inhibitor Cocktail, and 1 mM phenylmethanesulfonyl fluoride (PMSF). Cells were lysed by douncing on ice and lysates were clarified by ultracentrifugation and filtered through a 0.45 uM low protein-binding filter. Filtered supernatant was bound to a 5 ml StrepTrap HP column pre-equilibrated in buffer. The column was washed with buffer (30 ml), buffer supplemented with 5 mM ATP (30 ml), and a final wash with buffer (30 ml). Protein complexes were eluted in buffer supplemented with 2.5 mM d-Desthiobiotin. Pooled fractions were concentrated to ~500 ul in a Vivaspin 20 protein concentrator and loaded onto a Superose 6 Increase 10/300 column pre-equilibrated with buffer. Pooled fractions were concentrated to ~0.5–1 mg/ml in a Vivaspin 20 protein concentrator supplemented with 20% glycerol and flash frozen in liquid nitrogen.

## dsRNA preparation

106 nucleotide (nt) RNAs were prepared as described (*Sinha et al., 2018*; *Sinha et al., 2015*). 52nt and 42 nt RNAs were chemically synthesized by University of Utah DNA/RNA Synthesis Core or IDT. Equimolar amounts of ssRNAs were annealed in annealing buffer (50 mM TRIS pH 8.0, 20 mM KCl) by placing the reaction on a heat block (95 °C) and slow cooling ≥2 hr.

## dsRNA sequences

106 nt sense RNA: 5'-GGCAAUGAAAGACGGUGAGCUGGUGAUAUGGGAUAGUGUUCACCC UUGUUACACCGUUUUCCAUGAGCAAACUGAAACGUUUUCAUCGCUCUGGAGUGAAU ACCAA-3'.

106 nt antisense BLT RNA: 5'-UUGGUAUUCACUCCAGAGCGAUGAAAACGUUUCAGUU UGCUCAUGGAAAACGGUGUAACAAGGGUGAACACUAUCCCAUAUCACCAGCUCACC GUCUUUCAUUGCC-3'.

106 nt antisense 3'ovr RNA: 5'-GGUAUUCACUCCAGAGCGAUGAAAACGUUUCAGUUUG CUCAUGGAAAACGGUGUAACAAGGGUGAACACUAUCCCAUAUCACCAGCUCACCGU CUUUCAUUGCCAA-3'.

52 nt sense RNA: 5'-GGAGGUAGUAGGUUGUAUAGUAGUAAGACCAGACCCUAGACCAAU UCAUGCC-3'.

52 nt antisense BLT RNA: 5'-GGCAUGAAUUGGUCUAGGGUCUGGUCUUACUACUAUACAAC CUACUACCUCC-3'.

42 nt sense RNA: 5'-GGGAAGCUCAGAAUAUUGCACAAGUAGAGCUUCUCGAUCCCC-3'.

42 nt antisense BLT RNA: 5'-GGGGAUCGAGAAGCUCUACUUGUGCAAUAUUCUGAGCUUCC C-3'.

## Cleavage assays

Reactions were at 20 °C for times indicated in cleavage buffer (25 mM HEPES [pH 7.5], 50 mM KCl, 10 mM Mg(OAc)$_2$, 1 mM TCEP) with protein and dsRNA as specified,±5 mM ATP-MgOAc$_2$ or ATPγS-MgOAc$_2$. Reactions were started by adding indicated protein complex and stopped with 2 volumes of 2 X formamide loading buffer (95% formamide, 18 mM EDTA, 0.025% SDS, xylene cyanol, bromophenol blue). Products were separated on an 8 M Urea 17% denaturing PAGE gel, visualized on a PhosphorImager screen, and quantified with ImageQuant software. (Cleaved dsRNA / total dsRNA) x 100 = % dsRNA cleaved. Cleaved dsRNA is defined as all RNA products below the 106nt band. Total dsRNA is defined as the 106nt band plus all RNA products below the 106nt band. GraphPad Prism version 9 was used for curve-fitting analysis. Data were fit to the pseudo-first-order equation $y=y_o + A \times (1-e^{-kt})$; where y=product formed (% dsRNA cleaved); A=amplitude of the rate curve, $y_o$ = baseline (~0), k=pseudo-first-order rate constant = $k_{obs}$; t=time. Dotted line in graphs refers to the plateau constrained to 81.02 to compare the efficiency of all cleavage reactions to the wildtype complex with BLT dsRNA and ATP. For reactions with hexokinase and glucose, hexokinase (1 or 2 units as indicated) and glucose (10 mM) were incubated in reaction mix containing ±5 mM ATP (20 min; 20 C) to deplete ATP (or contaminating ATP) before adding DCR-1•DRH-1•RDE-4.

## ATP hydrolysis assays

Reactions were at 20 °C for times indicated in cleavage buffer with 25 nM protein and 400 nM dsRNA as specified, and 100 uM ATP-MgOAc$_2$ plus 100 nM [α-$^{32}$P] ATP-MgOAc$_2$ (3000 Ci/mmol) to monitor hydrolysis. Reactions were started by adding protein, stopped at indicated times in equal volume of 0.5 M EDTA, spotted onto PEI-cellulose plates, and chromatographed with 0.75 M KH$_2$PO$_4$ (adjusted to pH 3.3 with H$_3$PO$_4$) until solvent reached top of plate. Plates were dried and visualized on a PhosphorImager screen. ADP and ATP radioactivity signals in TLC plates were quantified with ImageQuant software. (ADP / (ADP +ATP)) x 100 = % ADP product formed. GraphPad Prism version 9 was used for curve-fitting analysis. Data were fit to the pseudo-first-order equation as described in cleavage assays quantification. Dotted line in graphs refers to the plateau constrained to 89.55 to compare the efficiency of all cleavage reactions to the wildtype complex with BLT dsRNA.

## Gel mobility-shift assays

Gel mobility shift assays were performed with 20pM 106-basepair BLT or 3'ovr dsRNA, with sense strand labeled with $^{32}$P at the 5' terminus and a 2', 3'-cyclic phosphate. Labeled dsRNA was incubated and allowed to reach equilibrium (45 min, 4°C) with wildtype DCR-1•DRH-1•RDE-4 complex, or indicated variant complex, in the presence and absence of 5 mM ATP-MgOAc$_2$, in binding buffer (25 mM HEPES pH 8.0, 100 mM KCl, 10 mM MgCl$_2$, 10% (vol/vol) glycerol, 1 mM TCEP); final reaction volume, 20 $\mu$l. Protein complex was serially diluted in binding buffer before addition to binding reaction.

Reactions were stopped by loading directly onto a 4% polyacrylamide (29:1 acrylamide/bisacrylamide) native gel running at 200 V at 4°C, in 0.5 X Tris/Borate/EDTA running buffer. The gel was pre-run (30 min) before loading samples. Gels were electrophoresed (3 hr) to resolve complex-bound dsRNA from free dsRNA, dried (80°C, 1 hr) and exposed overnight to a Molecular Dynamics Storage Phosphor Screen. Radioactivity signal was visualized on a Typhoon PhosphorImager (GE Healthcare Life-Sciences) in the linear dynamic range of the instrument and quantified using ImageQuant version 8 software. Radioactivity in gels corresponding to $dsRNA_{total}$ and $dsRNA_{free}$ was quantified to determine fraction bound. Fraction bound = 1 – ($dsRNA_{free}$ / $dsRNA_{total}$). All dsRNA that migrated through the gel more slowly than $dsRNA_{free}$ was considered as bound to the complex. To determine $K_d$ values, binding isotherms were fit using the Hill formalism, where fraction bound = $1/(1 + (K_d^n/[P]^n))$; $K_d$=equilibrium dissociation constant, n=Hill coefficient, [P]=protein concentration. GraphPad Prism version 9 was used for curve-fitting analysis.

### Trap assays
Reactions were at 20 °C for times indicated in cleavage buffer (25 mM HEPES [pH 7.5], 50 mM KCl, 10 mM Mg(OAc)$_2$, 1 mM TCEP) with 50 nM protein and 1 nM $^{32}$P internally labeled 106 BLT dsRNA with 5 mM ATP ±2000 nM 52 BLT cold dsRNA. Reactions were started by adding dsRNA and stopped with two volumes of 2 X formamide loading buffer (95% formamide, 18 mM EDTA, 0.025% SDS, xylene cyanol, bromophenol blue). Products were separated on an 8 M Urea 17% denaturing PAGE gel, visualized on a PhosphorImager screen, and quantified with ImageQuant software. (siRNA/total counts) x 100 = % siRNA product formed. GraphPad Prism version 9 was used for curve-fitting analysis. Data were fit to the pseudo-first-order equation as described in cleavage assays quantification.

### Cryo-EM specimen preparation
UltrAuFoil R2/2 Au300 mesh grids (Quantifoil) were glow discharged for 30 seconds on each side at 25 mA using a Pelco easiGlow unit (Ted Pella, Inc). A total of 3.5 µl of purified complex (0.8–1 µM concentration) with 1.5 x molar excess dsRNA with or without 5 mM nucleotide (ATP or ATPγS) and 5% glycerol was applied to the grid and blotted with filter paper (595 Filter Paper, Ted Pella, Inc) for 5 s using an Mk. II Vitrobot (Thermo Fisher Scientific) with a −1 mm offset and then plunge frozen into liquid ethane. For the sample that led to the DRH-1 bound to dsRNA reconstruction, the complex was purified as described above, except with 2 mM CaCl2 instead of 2 mM Mg(OAc)$^2$. 2 mM Mg(OAc)$^2$ was added to the sample before it was applied to the grid.

### Cryo-EM and data processing of complex bound to two dsRNAs
For the complex bound to two dsRNAs, cryo-EM movies were recorded using SerialEM v3.840 (*Mastronarde, 2005*) at the Pacific Northwest CryoEM Center. Images were recorded in super-resolution mode on a 300 kV Titan Krios (Thermo Fisher Scientific) equipped with a post-GIF K3 direct detector (Gatan, Inc), using a nominal magnification of 81,000 x, corresponding to a super-resolution pixel size of 0.394 Å, with a total dose of 50 electrons/Å$^2$ and 50 frames per movie.

Super-resolution cryo-EM movie frames were motion corrected, dose-weighted, Fourier binned 2 x, and summed using CryoSPARC v3.3.233 (*Punjani et al., 2017*). CTF parameters were determined using patch CTF estimation. A total of 3866 micrographs displayed CTF resolution fits of 6 Å or better and were used for downstream analysis. Particles were blob picked with a range of 225–375 Å diameter to obtain initial 2D classes for template-based picking. Particles were extracted at a box size of 300. During earlier processing steps, the box was 4 x binned but was extracted at the original box size of 300 for final reconstructions. After multiple rounds of template-based picking, a total of 324,464 particles were obtained, through which 70,904 of the best particles were selected and run through ab initio reconstruction. The best volume comprised 26,879 particles and revealed two dsRNA molecules interacting with the complex. A final homogeneous refinement job yielded a 7.6 Å reconstruction map that was used for rigid-body fitting of individual components.

### Cryo-EM and data processing of complex bound to one dsRNA
For the dataset yielding the reconstruction of the complex bound to one dsRNA, cryo-EM movies were recorded at a nominal magnification of 81,000×, corresponding to a super resolution pixel size of 0.533 Å, with a total dose of 50 electrons/Å$^2$ and 40 frames per movie. Data were collected using

Leginon (*Suloway et al., 2005*) at the National Center for CryoEM Access and Training (New York Structural Biology Center). Super-resolution cryo-EM movie frames were motion corrected, dose-weighted, Fourier binned 2 x, and summed using CryoSPARC v3.3.233 (*Punjani et al., 2017*).

Two separate datasets were collected of this complex, one with and one without added ATPγS. Separate processing of each dataset did not reveal noticeable differences, thus rationalizing the combining of the particles into one project. A total of 24,705 micrographs displaying CTF resolution fits of 6 Å or better were used for downstream analysis. Particles were selected from a combination of blob and template-based picking, then extracted at a box size of 384 pixels and 4 x binned for initial 2D classification. After several rounds of template-based picking from original blob-based picks (225–300 Å), 317,291 particles were used for 2D classification, and 314,554 particles were used for ab initio reconstruction. Unbinned particles (384 pixel box size) were used for final reconstructions. The volume with the best density for the entire complex with dsRNA bound (171,304 particles) were used for homogeneous and local refinement, yielding a final reconstruction at 6.1 Å resolution.

## Cryo-EM and data processing of DRH-1 bound to dsRNA

For the reconstruction of the DRH-1 dataset with calcium, magnesium, the complex, and 3'ovr 106 dsRNA, cryo-EM movies were recorded at a nominal magnification of 81,000 x, corresponding to a super-resolution pixel size of 0.529 Å with a total dose of 38 electrons/$Å^2$ and 40 frames per movie.

Movies were patch motion corrected and CTF estimated in CryoSPARC v3.3.233 (*Punjani et al., 2017*), and a total of 14,602 micrographs displaying CTF resolution fits of 6 Å or better were used for downstream analysis. Particles were selected using blob-based picking (225–300 Å diameter). After several rounds of 2D classification, a total of 759,994 were used for ab initio reconstruction (four classes). After removing junk classes, a total of 591,250 particles were used for heterogeneous refinement (four classes). This led to two similar, well-resolved classes comprising 362,869 particles that were then used for high-resolution 3D reconstruction. The particles were first subject to non-uniform refinement, producing a 3.1 Å map. Finally, local refinement of the same particles was performed using a mask that focused on DRH-1 density. This produced a final 2.9 Å map that was used for model building and refinement.

Data processing workflows for all three reconstructions are available in *Figure 4—figure supplements 1F and 2F* and *Supplementary file 1b*.

## Interpretation of DCR-1::DRH-1::RDE4::dsRNA reconstructions

For the densities for DCR-1 and RDE-4, we used the human model for rigid body fitting (PDB 5ZAK) (*Liu et al., 2018*). We also used the human model for rigid body fitting of DCR-1::RDE::dsRNA (PDB 5ZAL) (*Liu et al., 2018*). In both reconstructions of the complex, we used the AlphaFold2 (*Jumper et al., 2021*; *Varadi et al., 2022*) model to fit DRH-1 (AF-G5EDI8-F1). The NTD and helicase domains of DRH-1 were fit separately to accommodate the different orientation of the NTD in the AlphaFold2 model compared to the density. We generated an A-form dsRNA in UCSF Chimera (*Pettersen et al., 2004*) to fit into the density of the dsRNA interacting with DRH-1's helicase.

## Model building, refinement, and validation of DRH-1:dsRNA structure

The model for DRH-1 was built manually in the 2.9 Å density map using Coot v0.8.9.1 (*Emsley et al., 2010*). The AlphaFold2 (*Jumper et al., 2021*; *Varadi et al., 2022*) model as a starting point (AF-G5E-DI8-F1). The ADP ligand was built using Phenix (*Adams et al., 2010*) Ligand Fit tool, while the $Mg^{2+}$ and $Zn^{2+}$ substrates were manually built in Coot. We used a 106nt dsRNA in the sample; however, the density of dsRNA in the final reconstruction was shorter than 106 bp and likely represents a mixture of positions bound by DRH-1. We therefore generated a 30-nucleotide A:U base paired dsRNA in Coot to use for modeling. The model and substrates were subjected to real-space refinement using Phenix v1.20.1–4487. Default settings were used, except the weight was set to 0.01.

## Analytical ultracentrifugation

Sedimentation velocity analytical ultracentrifugation (Beckman Coulter Optima XL-I) of DCR-1•DRH-1•RDE-4 and DCR-1•DRH-1 after purification by size exclusion chromatography in 25 mM HEPES pH 7.5, 10 mM KOAc, 2 mM Mg(OAc)₂, 100 mM KCl, 1 mM TCEP. Samples (420 μL) at 0.4 A 280 nm (for DCR-1•DRH-1•RDE-4) and 0.3 A 280 nm (for DCR-1•DRH-1) and matching buffer (440 μL) were

centrifuged at 40,000 RPM in carbon epon 2-sector centerpieces loaded in an An60-Ti rotor. Radial distributions were recorded continuously via absorbance at 280 nm (0.003 radial step size, 1 replicate) and 20 °C. Buffer density ($\rho$) (1.005 g/mL), monomeric molecular weights (386,825 Da for DCR-1•DRH-1•RDE-4 and 343,417 Da for DCR-1•DRH-1), and partial specific volumes (0.7348 mL/g for DCR-1•DRH-1•RDE-4 and 0.735 mL/g for DCR-1•DRH-1) were calculated with SEDNTERP (*Philo, 2023*). The c(s) continuous sedimentation distribution function of the data was determined via analysis by SEDFIT using alternating Marquardt-Levenberg and simplex algorithms, with an F-ratio of 0.68, representing one standard deviation (*Schuck, 2000*). The c(s) analysis and data/fit/residuals were exported from SEDFIT and plotted with the GUSSI software (*Brautigam, 2015*).

## Materials availability

All materials are available upon reasonable request to the lead contact, Brenda Bass (bbass@biochem.utah.edu).

## Acknowledgements

We thank the Bass and Shen labs for helpful feedback, M Salemi at UC Davis Proteomics Core for assistance with mass spectrometry, S Wang for assistance with structural biology methodology, and T Duchaine for providing RDE-4 antibodies. DNA synthesis and flow cytometry (Office of The Director of the NIH, S10OD026959) was performed at the University of Utah Core Facilities. For the cryo-EM work, we acknowledge D Belnap at the University of Utah Electron Microscopy Core Laboratory, O Davulcu PNCC at OHSU (NIH grant U24GM129547), and H Kuang at NYSBC (NIH Common Fund Transformative High Resolution Cryo-Electron Microscopy program U24 GM129539, and by grants from the Simons Foundation, SF349247, and NY State Assembly). PNCC data was accessed through EMSL (grid.436923.9), a DOE Office of Science User Facility sponsored by the Office of Biological and Environmental Research and data for PNCC was accessed through EMSL, and data for NYSBC was accessed through Globus. We also thank I Allen and A Orendt at the Utah Center for High-Performance Computing for computational support. CDC was supported by the National Institutes of Health under Ruth L Kirschstein National Research Service Award NIH T32AI055434 from the National Institute of Allergy and Infectious Diseases (NIAID). This work was supported by funding to BLB from the National Institute of General Medical Sciences (R35GM141262) and the National Cancer Institute of the National Institutes of Health (R01CA260414) and to PSS from the National Institute of General Medical Sciences (R35GM133772).

## Additional information

### Funding

| Funder | Grant reference number | Author |
| --- | --- | --- |
| National Institute of General Medical Sciences | R35GM141262 | Brenda L Bass |
| National Cancer Institute | R01CA260414 | Brenda L Bass |
| National Institute of General Medical Sciences | R35GM133772 | Peter S Shen |
| National Institutes of Health | T32AI055434 | Claudia D Consalvo |

The funders had no role in study design, data collection and interpretation, or the decision to submit the work for publication.

### Author contributions

Claudia D Consalvo, Conceptualization, Data curation, Formal analysis, Validation, Investigation, Visualization, Methodology, Writing – original draft, Writing – review and editing; Adedeji M Aderounmu, Conceptualization, Formal analysis, Methodology, Writing – original draft, Writing – review and editing; Helen M Donelick, Methodology, Writing – original draft, Writing – review and editing; P

Joseph Aruscavage, Debra M Eckert, Methodology; Peter S Shen, Resources, Data curation, Supervision, Funding acquisition, Validation, Visualization, Methodology, Writing – review and editing; Brenda L Bass, Conceptualization, Resources, Formal analysis, Supervision, Funding acquisition, Writing – original draft, Project administration, Writing – review and editing

**Author ORCIDs**
Claudia D Consalvo ⓘ https://orcid.org/0000-0003-2038-4315
Peter S Shen ⓘ http://orcid.org/0000-0002-6256-6910
Brenda L Bass ⓘ http://orcid.org/0000-0003-1728-2254

Reviewer #1 (Public Review): https://doi.org/10.7554/eLife.93979.3.sa1
Reviewer #2 (Public Review): https://doi.org/10.7554/eLife.93979.3.sa2
Author response https://doi.org/10.7554/eLife.93979.3.sa3

## Additional files

**Supplementary files**
• Supplementary file 1. Supplemental tables. (a) Summary of $K_d$ values. (b) Cryo-EM data collection, refinement, and validation statistics.
• MDAR checklist

**Data availability**
CryoEM densities were deposited in EMDB under accession codes EMD-41060, EMD-43430, and EMD-43431. The DRH-1 model was deposited in PDB under accession code 8T5S. All codes are listed in *Supplementary file 1*b. Source data files have been provided for all gels displayed in this manuscript.

The following datasets were generated:

| Author(s) | Year | Dataset title | Dataset URL | Database and Identifier |
|---|---|---|---|---|
| Consalvo CD, Donelick HM, Shen PS, Brenda BL | 2024 | Cryo-EM structure of DRH-1 helicase and C-terminal domain bound to dsRNA | https://www.rcsb.org/structure/8T5S | RCSB Protein Data Bank, 8T5S |
| Consalvo CD, Donelick HM, Shen PS, Brenda BL | 2024 | Cryo-EM structure of *C. elegans* antiviral complex Dicer-1, DRH-1, and RDE-4 bound to one dsRNA | https://www.ebi.ac.uk/emdb/EMD-43430 | EMDataBank, EMD-43430 |
| Consalvo CD, Donelick HM, Shen PS, Brenda BL | 2024 | Cryo-EM structure of *C. elegans* antiviral complex Dicer-1, DRH-1, and RDE-4 bound to two dsRNA | https://www.ebi.ac.uk/emdb/EMD-43431 | EMDataBank, EMD-43431 |
| Consalvo CD, Donelick HM, Shen PS, Brenda BL | 2024 | Cryo-EM structure of DRH-1 helicase and C-terminal domain bound to dsRNA | https://www.ebi.ac.uk/emdb/EMD-41060 | EMDataBank, EMD-41060 |

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
