## [Editor Report · eLife assessment]

To investigate the evolutionary relationship between the RNAi pathway and innate immunity, this **valuable** study uses biochemistry and structural biology to investigate the trimeric complex of Dicer-1, DRH-1 (a RIG-I homologue), and RDE-4, which exists in *C. elegans*. The results described include rigorous kinetic analysis of the enzymatic activity of the complex and a moderate resolution cryo-EM structure. The results are **convincing** and add to the broader understanding of the evolution of antiviral defense.

---

## [Referee Report · Reviewer #1 (Public Review)]

Summary:

The authors establish a recombinant insect cell expression and purification scheme for the antiviral Dicer complex of *C. elegans*. In addition to Dicer-1, the complex harbors two additional proteins, the RIG-I-like helicase DRH-1 and the dsRNA-binding protein RDE-4. The authors show that the complex prefers blunt-end dsRNA over dsRNAs that contain overhangs. Furthermore, whereas ATP-dependent dsRNA cleavage only exacerbates regular dsRNA cleavage activity, the presence of RDE-4 is essential to ATP-dependent and ATP-independent dsRNA cleavage. Single-particle cryo-EM studies of the ternary *C. elegans* Dicer complex reveal that the N-terminal domain of DRH-1 interacts with the helicase domain of DCR-1, thereby relieving its autoinhibitory state. Last, the authors show that the ternary complex is able to processively cleave long dsRNA, an activity primarily relying on the helicase activity of DRH-1.

Strengths:

• First thorough biochemical characterization of the antiviral activity of *C. elegans* Dicer in complex with the RIG-I like helicase DRH-1 and the dsRNA-binding protein RDE-4

• Discovery that RDE-4 is essential to dsRNA processing, whereas ATP hydrolysis is not

• Discovery of an autoinhibitory role of DRH-1's N-terminal domain (in analogy to the CARD domains of RIG-I)

• First structural insights into the ternary complex DCR-1:DRH-1:RDE-4 by cryo-EM to medium resolution

• Trap experiments reveal that the ternary DCR-1 complex cleaves blunt-ended dsRNA processively. Likely, the helicase domain of DRH-1 is responsible for this processive cleavage.

Weaknesses:

• Cryo-EM Structure of the ternary Dicer-1:DRH-1:RED-4 complex to only medium resolution

• High-resolution structure of the C-terminal domain of DRH-1 bound to dsRNA does not reveal the mechanism of how blunt-end dsRNA and overhang-containing one are being discriminated

• The cryo-EM structure of DCR1:DRH-1:RDE-4 in the presence of ATP only reveals the helicase and CTD domains of DRH-1 bound to dsRNA. No information on dsRNA termini recognition is presented. The paragraph seems detached from the general flow of the manuscript.

---

## [Referee Report · Reviewer #2 (Public Review)]

Summary:

To investigate the evolutionary relationship between the RNAi pathway and innate immunity, this study uses biochemistry and structural biology to investigate the trimeric complex of Dicer-1, DRH-1 (a RIGI homologue) and RDE-4 , which exists in *C. elegans*. The three subunits were co-expressed to promote stable purification of the complex. This complex promoted ATP-dependent cleavage of blunt-ended dsRNAs. A detailed kinetic analysis was also carried out to determine the role of each subunit of the trimeric complex in both the specificity and efficiency of cleavage. These studies indicate that RDE-4 is critical for cleavage while DRC-1 is primarily involved in the specificity of the reaction, and DRH-1 promotes ATP hydrolysis. Finally, a moderate density (6-7 angstrom) cryo-EM structure of the trimeric complex is provided.

Strengths:

(1) Newly described methods for studying the *C. elegans* DICER complex

(2) New structure, albeit only moderate resolution

(3) Kinetic study of the complex in the presence and absence of individual subunits and mutations, provide detailed insight into the contribution of each subunit

Weaknesses:

(1) Limited insight due to limited structural resolution.

---

## [Author Response]

The following is the authors’ response to the original reviews.

**Public Reviews:**

**Reviewer #1 (Public Review):**
Summary:The authors establish a recombinant insect cell expression and purification scheme for the antiviral Dicer complex of *C. elegans*. In addition to Dicer-1, the complex harbors two additional proteins, the RIG-I-like helicase DRH-1, and the dsRNA-binding protein RDE-4. The authors show that the complex prefers blunt-end dsRNA over dsRNAs that contain overhangs. Furthermore, whereas ATP-dependent dsRNA cleavage only exacerbates regular dsRNA cleavage activity, the presence of RDE-4 is essential to ATP-dependent and ATP-independent dsRNA cleavage. Single-particle cryo-EM studies of the ternary *C. elegans* Dicer complex reveal that the N-terminal domain of DRH-1 interacts with the helicase domain of DCR-1, thereby relieving its autoinhibitory state. Lastly, the authors show that the ternary complex is able to processively cleave long dsRNA, an activity primarily relying on the helicase activity of DRH-1.Strengths:First thorough biochemical characterization of the antiviral activity of *C. elegans* Dicer in complex with the RIG-I-like helicase DRH-1 and the dsRNA-binding protein RDE-4. • Discovery that RDE-4 is essential to dsRNA processing, whereas ATP hydrolysis is not.Discovery of an autoinhibitory role of DRH-1's N-terminal domain (in analogy to the CARD domains of RIG-I).First structural insights into the ternary complex DCR-1:DRH-1:RDE-4 by cryo-EM to medium resolution.Trap experiments reveal that the ternary DCR-1 complex cleaves blunt-ended dsRNA processively. Likely, the helicase domain of DRH-1 is responsible for this processive cleavage.

We thank the reviewer for this accurate and thoughtful summary of the strengths of our study. We note that although ATP hydrolysis is not essential for dsRNA processing, it is essential for promoting an alternative, and dramatically more efficient, cleavage mechanism that is wellsuited for processing viral dsRNA.

Weaknesses:Cryo-EM Structure of the ternary Dicer-1:DRH-1:RED-4 complex to only medium resolution.

We agree with the reviewer that our structures are only of modest resolution. We continue to work towards a higher resolution structure of this conformationally heterogeneous complex. We do want to emphasize that despite our modest resolution, our structures provide novel insights into how the factors in the antiviral complex interact with each other, and also allow us to compare our findings to other Dicer systems. For example, the dsRNA binding protein RDE-4 binds the Hel2i subdomain, and this is similar to accessory dsRNA binding proteins of other Dicers, including human and *Drosophila*. Most importantly, for the first time, we uncover the interaction of DRH-1 with *C. elegans* Dicer; our structures show DRH-1's N-terminal domain interacting with Dicer's helicase domain. This observation spurred our experiments that showed the N-terminal domain of DRH-1, like the analogous domain of RIG-I, enables an autoinhibited conformation. While RIG-I autoinhibition is relieved by dsRNA binding, we do not observe this with *C. elegans* DRH-1 and speculate that instead it is the interaction with Dicer's helicase domain that relieves autoinhibition.

High-resolution structure of the C-terminal domain of DRH-1 bound to dsRNA does not reveal the mechanism of how blunt-end dsRNA and overhang-containing one are being discriminated.The cryo-EM structure of DCR1:DRH-1:RDE-4 in the presence of ATP only reveals the helicase and CTD domains of DRH-1 bound to dsRNA. No information on dsRNA termini recognition is presented. The paragraph seems detached from the general flow of the manuscript.

We agree with the reviewer that our paper would be improved with a high-resolution structure of DRH-1 bound to the dsRNA terminus to better understand terminus discrimination. Since we did not obtain a high-resolution structure of DRH-1 bound to the dsRNA terminus, we could not comment on how DRH-1 discriminates termini. However, our structure of DRH-1’s helicase and CTD bound to the middle of the dsRNA does provide important evidence that DRH-1 translocates along dsRNA, which is crucial for our interpretation of DRH-1’s ATPase function in the antiviral complex. Furthermore, our analysis of the DRH-1:dsRNA contacts reveals just how well conserved DRH-1 is with mammalian RLRs.

The antiviral DCR-1:DRH-1:RDE-4 complex shows largely homologous activities and regulation than *Drosophila* Dicer-2.

It is unclear to us why this is a weakness. In our Discussion in the section “Relationship to previously characterized Dicer activities,” we compare and contrast the *C. elegans* antiviral complex and the most well characterized antiviral Dicer: *Drosophila* Dcr2. While it might not be surprising that two invertebrate activities that both must target viral dsRNA have similar enzymatic properties, we find this remarkable given that Dcr2 orchestrates cleavage with a single protein, while two helicases and a dsRNA binding protein cooperate in the *C. elegans* reaction. Our careful biochemical analyses reveal how the three proteins cooperate. In vivo, *C. elegans* Dicer must function to cleave pre-miRNAs, endogenous siRNAs as well as viral dsRNA, and we speculate that the use of diverse accessory factors allows *C. elegans* Dicer to carry out these distinct tasks.

**Reviewer #2 (Public Review):**
Summary:To investigate the evolutionary relationship between the RNAi pathway and innate immunity, this study uses biochemistry and structural biology to investigate the trimeric complex of Dicer1, DRH-1 (a RIGI homologue), and RDE-4, which exists in *C. elegans*. The three subunits were co-expressed to promote stable purification of the complex. This complex promoted ATPdependent cleavage of blunt-ended dsRNAs. A detailed kinetic analysis was also carried out to determine the role of each subunit of the trimeric complex in both the specificity and efficiency of cleavage. These studies indicate that RDE-4 is critical for cleavage while DRC-1 is primarily involved in the specificity of the reaction, and DRH-1 promotes ATP hydrolysis. Finally, a moderate density (6-7 angstrom) cryo-EM structure is presented with attempts to position each of the components.Strengths:(1) Newly described methods for studying the *C. elegans* DICER complex.(2) New structure, albeit only moderate resolution.(3) Kinetic study of the complex in the presence and absence of individual subunits and mutations, provides detailed insight into the contribution of each subunit.Weaknesses:(1) Limited insight due to limited structural resolution.(2) No attempts to extend findings to other Dicer or RLR systems.

Overall, we agree with the assessment of this reviewer, and we thank them for their efforts in evaluating our manuscript. Whenever possible we have discussed the similarities and differences of the *C. elegans* Dicer to other Dicers and RLR systems. We are unsure how we could have expanded upon this further (as suggested in point 2).

**Recommendations for the authors:**

**Reviewer #1 (Recommendations For The Authors):**
Minor recommendations to the authors:Page 10: To assess the role of ATP hydrolysis for dsRNA binding, please refrain from using the term "fuzzy band" as a qualitative measure of RNA binding to the ternary complexes.

We searched our entire manuscript and did not find the term “fuzzy band.” We did describe some of the bands in the gel shift assays as “diffuse.” This is an accurate description of the bands we see in our gels and distinguishes them from other more well-defined bands.

Page 13: "positioned internally" - please explain "internally" better here.

We agree with the reviewer that “positioned internally” is confusing. In our revised manuscript we have changed this sentence to (Page 13, line 1):

“Under these conditions, we obtained a 2.9 Å reconstruction of the helicase and CTD domains of DRH-1 bound to the middle region of the dsRNA, rather than its terminus (Figures 4C and S9), suggesting that DRH-1 hydrolyzes ATP to translocate along dsRNA.”

Page 13: Please re-consider the detailed description of the dsRNA:DRH-1 contacts.

We feel it is very important to illustrate and describe these contacts, which will be of interest to those who study mammalian RLRs.

Figure 1C/D: Please write "minus/+ ATP" on top of the gels to make this distinction more clearly visible.

In our original manuscript the gels are labeled with “minus ATP” (panel C) or “5mM ATP” (panel D) on the left to indicate both gels in panel C and both gels in panel D have the same conditions. This is also stated in the figure legend. We have not made revisions in response to this comment because we think it is already clear.

Figure 2: Please explain R = RDE-4 in a clearly visible legend.

We agree with the author that the illustration above the gels was not explained clearly. In our revised manuscript we have added the sentence below to the beginning of Figure legend 2A. “Cartoons indicate complexes and variants, with mutations in DCR-1 (green) and DRH-1 (blue) indicated with the amino acid change, and the presence of RDE-4 (R) represented with a purple circle.”

Figure 4A: Please label the DRH-1 helicase domain and the C-terminal domain.

We agree with the reviewer that we could more clearly define our labeled domains. In the revised manuscript we have added a sentence to the legend of Figure 4A: “The domains of DCR-1, DRH-1, and RDE-4 are color coded the same as in Fig 1A. For simplicity, only domains discussed in the text are labeled.”

**Reviewer #2 (Recommendations For The Authors):**
This study is complete in that all necessary controls and data are included and the authors are careful in their interpretation so as to not overstate the data or conclusions.The only suggestion is that further extension of the study to address the weaknesses above would increase the breadth of impact of this work.

We thank the reviewer for their thoughtful comments. Weaknesses are addressed above in public reviews, and we will add again that we agree that a higher resolution structure would provide additional insight. In ongoing research, we are working towards this goal.